# Distributed Newton-Type Methods with Communication Compression and Bernoulli Aggregation

**Rustem Islamov**                                          *rustem.islamov@ip-paris.fr*
*Institut Polytechnique de Paris*
*Palaiseau, France*

**Xun Qian**
*JD Explore Academy*
*Beijing, China*

**Slavomír Hanzely**
*King Abdullah University of Science and Technology*
*Thuwal, Saudi Arabia*

**Mher Safaryan**
*King Abdullah University of Science and Technology*
*Thuwal, Saudi Arabia*

**Peter Richtárik**
*King Abdullah University of Science and Technology*
*Thuwal, Saudi Arabia*

**Reviewed on OpenReview:** *https://openreview.net/forum?id=NekBTCKJ1H*

## Abstract

Despite their high computation and communication costs, Newton-type methods remain an appealing option for distributed training due to their robustness against ill-conditioned convex problems. In this work, we study *communication compression* and *aggregation mechanisms* for curvature information in order to reduce these costs while preserving theoretically superior local convergence guarantees. We show that the recently developed class of *three point compressors (3PC)* of (Richtárik et al., 2022) for gradient communication can be generalized to Hessian communication as well. This result opens up a wide variety of communication strategies, such as *contractive compression* and *lazy aggregation*, available to our disposal to compress prohibitively costly curvature information. Moreover, we discovered several new 3PC mechanisms, such as *adaptive thresholding* and *Bernoulli aggregation*, which require reduced communication and occasional Hessian computations. Furthermore, we extend and analyze our approach to bidirectional communication compression and partial device participation setups to cater to the practical considerations of applications in federated learning. For all our methods, we derive fast *condition-number-independent* local linear and/or superlinear convergence rates. Finally, with extensive numerical evaluations on convex optimization problems, we illustrate that our designed schemes achieve state-of-the-art communication complexity compared to several key baselines using second-order information.

## 1 Introduction

In this work we consider the distributed optimization problem given by the form of ERM:

$$\min_{x \in \mathbb{R}^d} \left\{ f(x) := \frac{1}{n} \sum_{i=1}^{n} f_i(x) \right\}, \tag{1}$$

where $d$ is the (potentially large) number of parameters of the model $x \in \mathbb{R}^d$ we aim to train, $n$ is the (potentially large) number of devices in the distributed system, $f_i(x)$ is the loss/risk associated with the data stored on machine $i \in [n] := \{1, 2, \ldots, n\}$, and $f(x)$ is the empirical loss/risk.

In order to jointly train a single machine learning model using all devices' local data, *collective efforts* are necessary from all compute nodes. Informally, each entity should invest some "knowledge" from its local "wisdom" to create the global "wisdom". The classical approach in distributed training to implement the collective efforts was to literally collect all the raw data devices acquired and then perform the training in one place with traditional methods. However, the mere access to the raw data hinders the clients' *data privacy* in federated learning applications (Konečný et al., 2016b;a; McMahan et al., 2017). Besides, even if we ignore the privacy aspect, accumulating all devices' data into a single machine is often *infeasible* due to its increasingly large size (Bekkerman et al., 2011).

Because of these considerations, there has been a serious stream of works studying distributed training with decentralized data. This paradigm of training brings its own advantages and limitations. Perhaps the major advantage is that each remote device's data can be processed simultaneously using *local computational resources*. Thus, from another perspective, we are scaling up the traditional single-device training to a distributed training of multiple parallel devices with decentralized data and local computation. However, the cost of scaling the training over multiple devices forces *intensive communication* between nodes, which is the *key bottleneck* in distributed systems.

## 1.1 Related work: from first-order to second-order distributed optimization

Currently, first-order optimization methods are the default options for large-scale distributed training due to their cheap per-iteration costs. Tremendous amount of work has been devoted to extend and analyze gradient-type algorithms to conform to various practical constraints such as efficient communication through compression mechanisms (Alistarh et al., 2017; 2018b; Wen et al., 2017; Wangni et al., 2018; Sahu et al., 2021; Tyurin & Richtárik, 2022) and local methods (Gorbunov et al., 2021b; Stich, 2020; Karimireddy et al., 2020; Nadiradze et al., 2021a; Mishchenko et al., 2022), peer-to-peer communication through graphs (Koloskova et al., 2019; 2020; Kovalev et al., 2021), asynchronous communication (Feyzmahdavian & Johansson, 2021; Nadiradze et al., 2021b), partial device participation (Yang et al., 2021), Byzantine or adversarial attacks (Karimireddy et al., 2021; 2022), faster convergence through acceleration (Allen-Zhu, 2017; Li et al., 2020b; Qian et al., 2021) and variance reduction techniques (Lee et al., 2017; Mishchenko et al., 2019; Horváth et al., 2019; Cen et al., 2020; Gorbunov et al., 2021a), data privacy and heterogeneity over the nodes (Kairouz et al, 2019; Li et al., 2020a),

Nevertheless, despite their wide applicability, all first-order methods (including accelerated ones) inevitably suffer from ill-conditioning of the problem. In the past few years, several algorithmic ideas and mechanisms to tackle the above-mentioned constraints have been adapted for second-order optimization. The goal in this direction is to enhance the convergence by increasing the resistance of gradient-type methods against ill-conditioning using the knowledge of curvature information. The basic motivation that the Hessian computation will be useful in optimization is the fast *condition-number-independent* (local) convergence rate of classic Newton's method (Beck, 2014), that is beyond the reach of *all* first-order methods.

Because of the quadratic dependence of Hessian information ($d^2$ floats per each Hessian matrix) from the dimensionality of the problem, the primary challenge of taming second-order methods was efficient communication between the participating devices. To alleviate prohibitively costly Hessian communication, many works such as DiSCO (Zhang & Xiao, 2015; Zhuang et al., 2015; Lin et al., 2014; Roosta et al., 2019), GIANT (Wang et al., 2018; Shamir et al., 2014; Reddi et al., 2016) and DINGO (Crane & Roosta, 2019; Ghosh et al., 2020b) impart second-order information by condensing it into Hessian-vector products. Inspired from compressed first-order methods, an orthogonal line of work, including DAN-LA (Zhang et al., 2020b), Quantized Newton (Alimisis et al., 2021), NewtonLearn (Islamov et al., 2021), FedNL (Safaryan et al., 2022), Basis Learn (Qian et al., 2022) and IOS (Fabbro et al., 2022), applies lossy compression strategies directly to Hessian matrices reducing the number of encoding bits. Other techniques that have been migrated from first-order optimization literature are local methods (Gupta et al., 2021), partial device participation (Safaryan et al., 2022; Qian et al., 2022), defenses against Byzantine attacks (Ghosh et al., 2020a;c). The

theoretical comparison of second order methods is presented in Table 1. We defer more detailed review of a literature of second-order methods to the Appendix.

**Table 1:** Theoretical comparison of several second order methods (including ours) in strongly convex setup with Lipschitz continuous Hessians. Advantages are written in green, while limitations are colored in red.

| Method | LipC grad[1] | Comm. Cost[2] | Comp. Cost[3] | Rate | Comments |
|---|---|---|---|---|---|
| GIANT[4] (Wang et al., 2018) | No | $\mathcal{O}(d)$ | Full Hessian | Local $\kappa$-dependent linear. Global $\mathcal{O}(\log \kappa/\epsilon)$, quadratics | Big data regime (#data $\gg d$) |
| DINGO[5,6] (Crane & Roosta, 2019) | No | $\mathcal{O}(d)$ | Hessian-vector products | Global linear, but no fast local | Also requires Hessian pseudo-inverse and vector products. |
| DAN (Zhang et al., 2020b) | No | $\mathcal{O}(nd^2)$ | Full Hessian | Global quadratic rate after $\mathcal{O}(L/\mu^2)$ iterations. | - |
| DAN-LA (Zhang et al., 2020b) | Yes | $\mathcal{O}(nd)$ | Full Hessian | Asymptotic and implicit global superlinear rate. | $\lim_{k \to \infty} \frac{\|x_{k+1}-x^*\|}{\|x_k-x^*\|} = 0$ Independent of $\kappa$ ? Better non-asymptotic complexity over linear rate ? |
| NL[4] (Islamov et al., 2021) | No | $\mathcal{O}(d)$ | Full Hessian | Local linear and superlinear independent of $\kappa$, but dependent on #data. global linear | reveals local data to server |
| Quantized Newton[6] (Alimisis et al., 2021) | Yes | $\widetilde{\mathcal{O}}(d^2)$ | Full Hessian | Local (fixed) linear without global | - |
| FedNL (Safaryan et al., 2022) | No | $\mathcal{O}(d)$ | Full Hessian | Local (fixed) linear and superlinear, independent of $\kappa$ and #data Global linear | Supports contractive Hessian compression, Bidirectional compression. |
| BL (Qian et al., 2022) | No | $\mathcal{O}(d)$ | Full Hessian | Local (fixed) linear and superlinear, independent of $\kappa$ and #data Global linear | Supports contractive Hessian compression, Bidirectional compression. Exploits lower intrinsic dimensionality of data. |
| Fib-IOS[6] (Fabbro et al., 2022) | Yes | $\mathcal{O}(d)$ | Periodic Full Hessian | implicit global linear | Only rank-type compression. Backtracking line search. SVD in each round. |
| FLECS (Agafonov et al., 2022) | Yes | $\mathcal{O}(d)$ | Hessian-vector products | Implicit global linear, but no fast local[7] | Backtracking line search. SVD in each round. |
| **Newton-3PC (this work)** | No | $\mathcal{O}(d)$ | Periodic Full Hessian and/or Hessian-vector products | Local (fixed) linear and superlinear, independent of $\kappa$, independent of #data. Global rate[8] | Supports contractive Hessian compression, Bidirectional compression. |

[1] **LipC grad** = **Lip**schitz **C**ontinuous **grad**ients.
[2] **Comm. Cost** = **Comm**unication **Cost** per round.    [3] **Comp. Cost** = **Comp**utation **Cost** per round.
[4] Only for Generalized Linear Models, e.g. $loss_j(x; a_j) = \phi_j(a_j^\top x) + \lambda\|x\|^2$.
[5] Uses Moral Smoothness: $\|\nabla^2 f(x)\nabla f(x) - \nabla^2 f(y)\nabla f(y)\| \le L\|x-y\|$.    [6] Strongly convex local loss functions for all clients.
[7] FLECS has a local rate under the condition that all iterates remain within some fixed neighborhood of the optimum.
[8] See Section I in the Appendix for globalization strategies.

## 2   Motivation and Contributions

Handling and taking advantage of the second-order information in distributed setup is rather challenging. As opposed to gradient-type methods, Hessian matrices are both harder to compute and much more expensive to communicate. To avoid directly accessing costly Hessian matrices, methods like DiSCO (Zhang & Xiao, 2015), GIANT (Wang et al., 2018) and DINGO (Crane & Roosta, 2019) exploit Hessian-vector products only, which are as cheap to compute as gradients (Pearlmutter, 1994). However, these methods typically suffer from data heterogeneity, need strong assumptions on problem structure (e.g., generalized linear models) and/or do not provide fast local convergence rates.

On the other hand, recent works (Safaryan et al., 2022; Qian et al., 2022) have shown that, with the access of Hessian matrices, fast local rates can be guaranteed for solving general finite sums (1) under compressed communication and arbitrary heterogeneous data. In view of these advantages, in this work we adhere to this approach and study communication mechanisms that can further lighten communication and reduce computation costs. Below, we summarize our key contributions.

## 2.1 Flexible communication strategies for Newton-type methods

We prove that the recently developed class of *three point compressors (3PC)* of Richtárik et al. (2022) for gradient communication can be generalized to Hessian communication as well. In particular, we propose a new method, which we call Newton-3PC (Algorithm 1), extending FedNL (Safaryan et al., 2022) algorithm for arbitrary 3PC mechanism. This result opens up a wide variety of communication strategies, such as *contractive compression* (Stich et al., 2018a; Alistarh et al., 2018a; Karimireddy et al., 2019) and *lazy aggregation* (Chen et al., 2018; Sun et al., 2019a; Ghadikolaei et al., 2021), available to our disposal to compress prohibitively costly curvature information. Besides, Newton-3PC (and its local convergence theory) recovers FedNL (Safaryan et al., 2022) (when *contractive compressors* are used as 3PC) and BL (Qian et al., 2022) (when *rotation compression* is used as 3PC) in special cases.

## 2.2 New compression and aggregation schemes

Moreover, we discovered several new 3PC mechanisms, which require reduced communication and occasional Hessian computations. In particular, to reduce communication costs, we design an *adaptive thresholding* (Example 3.3) that can be seamlessly combined with an already adaptive *lazy aggregation* (Example 3.7). In order to reduce computation costs, we propose *Bernoulli aggregation* (Example 3.9) mechanism which allows local workers to *skip* both computation and communication of local information (e.g., Hessian and gradient) with some predefined probability. Moreover, *sketch-and-project* operator (Example 3.5) reduces the computation costs relying on Hessian-vector products.

## 2.3 Extensions

Furthermore, we provide several extensions to our approach to cater to the practical considerations of applications in federated learning. In the main part of the paper, we consider only bidirectional communication compression (Newton-3PC-BC) setup, where we additionally apply Bernoulli aggregation for gradients (worker to server direction) and another 3PC mechanism for the global model (server to worker direction). The extension for partial device participation (Newton-3PC-BC-PP) setup and the discussion for globalization are deferred to the Appendix.

## 2.4 Fast local linear/superlinear rates

All our methods are analyzed under the assumption that the global objective is strongly convex and local Hessians are Lipschitz continuous. In this setting, we derive fast *condition-number-independent* local linear and/or superlinear convergence rates.

## 2.5 Extensive experiments and Numerical Study

Finally, with extensive numerical evaluations on convex optimization problems, we illustrate that our designed schemes achieve state-of-the-art communication complexity compared to several key baselines using second-order information.

# 3 Three Point Compressors for Matrices

To properly incorporate second-order information in distributed training, we need to design an efficient strategy to synchronize locally evaluated $d \times d$ Hessian matrices. Simply transferring $d^2$ entries of the matrix each time it gets computed would put significant burden on communication links of the system. Recently, Richtárik et al. (2022) proposed a new class of gradient communication mechanisms under the name *three point compressors (3PC)*, which unifies contractive compression and lazy aggregation mechanisms into one class. Here we extend the definition of 3PC for matrices under the Frobenius norm $\| \cdot \|_F$ and later apply to matrices involving Hessians.

**Definition 3.1** (3PC for Matrices). *We say that a (possibly randomized) map*

$$\mathcal{C}_{\mathbf{H},\mathbf{Y}}(\mathbf{X}) : \underbrace{\mathbb{R}^{d \times d}}_{\mathbf{H} \in} \times \underbrace{\mathbb{R}^{d \times d}}_{\mathbf{Y} \in} \times \underbrace{\mathbb{R}^{d \times d}}_{\mathbf{X} \in} \to \mathbb{R}^{d \times d} \tag{2}$$

*is a three point compressor (3PC) if there exist constants $0 < A \le 1$ and $B \ge 0$ such that*

$$\mathbb{E}\left[\|\mathcal{C}_{\mathbf{H},\mathbf{Y}}(\mathbf{X}) - \mathbf{X}\|_{\mathrm{F}}^2\right] \le (1 - A)\|\mathbf{H} - \mathbf{Y}\|_{\mathrm{F}}^2 + B\|\mathbf{X} - \mathbf{Y}\|_{\mathrm{F}}^2. \tag{3}$$

*holds for all matrices $\mathbf{H}, \mathbf{Y}, \mathbf{X} \in \mathbb{R}^{d \times d}$.*

The matrices $\mathbf{Y}$ and $\mathbf{H}$ can be treated as parameters defining the compressor that would be chosen adaptively. Once they fixed, $\mathcal{C}_{\mathbf{H},\mathbf{Y}} : \mathbb{R}^{d \times d} \to \mathbb{R}^{d \times d}$ is a map to compress a given matrix $\mathbf{X}$. Let us discuss special cases with some examples.

**Example 3.2** (**Contractive compressors** (Karimireddy et al., 2019)). *The (possibly randomized) map $\mathcal{C} : \mathbb{R}^{d \times d} \to \mathbb{R}^{d \times d}$ is called contractive compressor with contraction parameter $\alpha \in (0, 1]$, if the following holds for any matrix $\mathbf{X} \in \mathbb{R}^{d \times d}$*

$$\mathbb{E}\left[\|\mathcal{C}(\mathbf{X}) - \mathbf{X}\|_{\mathrm{F}}^2\right] \le (1 - \alpha)\|\mathbf{X}\|_{\mathrm{F}}^2. \tag{4}$$

Notice that (4) is a special case of (2) when $\mathbf{H} = \mathbf{0}$, $\mathbf{Y} = \mathbf{X}$ and $A = \alpha$, $B = 0$. Therefore, contractive compressors are already included in the 3PC class. Contractive compressors cover various well known compression schemes such as greedy sparsification, low-rank approximation and (with a suitable scaling factor) arbitrary unbiased compression operator (Beznosikov et al., 2020). There have been several recent works utilizing these compressors for compressing Hessian matrices (Zhang et al., 2020b; Alimisis et al., 2021; Islamov et al., 2021; Safaryan et al., 2022; Qian et al., 2022; Fabbro et al., 2022). Below, we introduce yet another contractive compressor based on thresholding idea which shows promising performance in our experiments.

**Example 3.3** (**Adaptive Thresholding [NEW]**). *Following Sahu et al. (2021), we design an* adaptive thresholding *operator with parameter $\lambda \in (0, 1]$ defined as follows; for all $j, l \in [d]$ and $\mathbf{X} \in \mathbb{R}^{d \times d}$*

$$[\mathcal{C}(\mathbf{X})]_{jl} := \begin{cases} \mathbf{X}_{jl} & \text{if } |\mathbf{X}_{jl}| \ge \lambda \|\mathbf{X}\|_\infty, \\ 0 & \text{otherwise,} \end{cases} \tag{5}$$

In contrast to *hard thresholding* operator of Sahu et al. (2021), (5) uses adaptive threshold $\lambda\|\mathbf{X}\|_\infty$ instead of fixed threshold $\lambda$. With this choice, we ensures that at least the Top-1 is transferred. In terms of computation, thresholding approach is more efficient than Top-$K$ (Stich et al., 2018b) as only single pass over the values is already enough instead of partial sorting.

**Lemma 3.4.** *The adaptive thresholding operator (5) is contractive with $\alpha = \max(1 - (d\lambda)^2, 1/d^2)$.*

A popular technique to decrease computation cost of Newton-type methods is to rely on Hessian-vector products. We show that so called *sketch-and-project* mechanism (Gower & Richtárik, 2017) is a special case of contractive compressor.

**Example 3.5** (**Sketch-and-Project** (Gower & Richtárik, 2017)). *Let $\mathbf{S}$ be a sketching matrix sampled from a fixed distribution $\mathcal{D}$ over matrices in $\mathbb{R}^{d \times \tau}$ ($\tau \ge 1$ can but does not need to be fixed). We define sketch-and-project operator as follows*

$$\mathcal{C}(\mathbf{X}) = \mathbf{S}(\mathbf{S}^\top \mathbf{S})^\dagger \mathbf{S}^\top \mathbf{X}. \tag{6}$$

For more details on sketch-and-project operator we refer a reader to the Appendix.

**Lemma 3.6.** *The sketch-and-project operator (6) is a contractive compressor with $\alpha = \lambda_{\min}^+(\mathbb{E}\left[\mathbf{S}(\mathbf{S}^\top \mathbf{S})^\dagger \mathbf{S}^\top\right])$ where the expectation is taken w.r.t. randomness of the sketching $\mathbf{S}$, and $\lambda_{\min}^+(\mathbf{M})$ indicates the smallest positive eigenvalue of a symmetric matrix $\mathbf{M}$.*

The next two examples are 3PC schemes which in addition to contractive compressors utilize aggregation mechanisms, which is an orthogonal approach to contractive compressors.

**Example 3.7** (**Compressed Lazy AGgregation (CLAG)** (Richtárik et al., 2022))**.** *Let $\mathcal{C}\colon \mathbb{R}^{d\times d} \to \mathbb{R}^{d\times d}$ be a contractive compressor with contraction parameter $\alpha \in (0,1]$ and $\zeta \geq 0$ be a trigger for the aggregation. Then CLAG mechanism is defined as*

$$\mathcal{C}_{\mathbf{H},\mathbf{Y}}(\mathbf{X}) = \begin{cases} \mathbf{H} + \mathcal{C}(\mathbf{X} - \mathbf{H}) & \textit{if } \|\mathbf{X} - \mathbf{H}\|_{\mathrm{F}}^2 > \zeta\|\mathbf{X} - \mathbf{Y}\|_{\mathrm{F}}^2 \\ \mathbf{H} & \textit{otherwise} \end{cases} \tag{7}$$

In the special case of identity compressor $\mathcal{C} = \mathrm{Id}$ (i.e., $\alpha = 1$), CLAG reduces to lazy aggregation (Chen et al., 2018). On the other extreme, if the trigger $\zeta = 0$ is trivial, CLAG recovers recent variant of error feedback for contractive compressors, i.e., EF21 mechanism (Richtárik et al., 2021).

**Lemma 3.8** (see Lemma 4.3 in (Richtárik et al., 2022))**.** *CLAG mechanism* (7) *is a 3PC compressor with $A = 1 - (1-\alpha)(1+s)$ and $B = \max\{(1-\alpha)(1+1/s), \zeta\}$, for any $s \in (0, \alpha/(1-\alpha))$.*

From the first glance, the structure of CLAG in (7) may not seem communication efficient as the the matrix $\mathbf{H}$ (appearing in both cases) can potentially by dense. However, as we will see in the next section, $\mathcal{C}_{\mathbf{H},\mathbf{Y}}$ is used to compress $\mathbf{X}$ when there is no need to communicate $\mathbf{H}$. Thus, with CLAG we either send compressed matrix $\mathcal{C}(\mathbf{X} - \mathbf{H})$ if the condition with trigger $\zeta$ activates or nothing.

**Example 3.9** (**Compressed Bernoulli AGgregation (CBAG)** [**NEW**])**.** *Let $\mathcal{C}\colon \mathbb{R}^{d\times d} \to \mathbb{R}^{d\times d}$ be a contractive compressor with contraction parameter $\alpha \in (0,1]$ and $p \in (0,1]$ be the probability for the aggregation. We then define CBAG mechanism is defined as*

$$\mathcal{C}_{\mathbf{H},\mathbf{Y}}(\mathbf{X}) = \begin{cases} \mathbf{H} + \mathcal{C}(\mathbf{X} - \mathbf{H}) & \textit{with prob. } p, \\ \mathbf{H} & \textit{with prob. } 1 - p. \end{cases} \tag{8}$$

The advantage of CBAG (8) over CLAG is that there is no condition to evaluate and check. This choice of probabilistic switching reduces computation costs as with probability $1 - p$ it is useless to compute $\mathbf{X}$. Note that CBAG has two independent sources of randomness: Bernoulli aggregation and possibly random operator $\mathcal{C}$.

**Lemma 3.10.** *CBAG mechanism* (8) *is a 3PC compressor with $A = (1-p\alpha)(1+s)$ and $B = (1-p\alpha)(1+1/s)$, for any $s \in (0, p\alpha/(1-p\alpha))$.*

Lazy aggregation communication mechanisms empirically outperform vanilla GD (Chen et al., 2018). The main idea is to communicate local gradients/Hessians only in the case when a certain conidition holds. For example, in the case of CLAG we update local Hessian estimators only if they are sufficiently far from true local Hessians. Parameter $\zeta$ controls how often we want to skip communication. If it is large, then we reuse previous Hessian estimators more often. CLAG can be seen as an adaptive mechanism that dynaimcally updates estimators based on the conditions that change throughout the optimization process. Thus, it is difficult to estimate/approximate how frequently we skip communications.

In fact, lazy aggregation remains poorly studied field in the literature. It has been analyzed for gradient-type methods only. However, the theoretical analysis show either explicit convergence guarantees (Sun et al., 2019b; Chen et al., 2018), or sublinear rates in the strongly convex regime (Shokri Ghadikolaei et al., 2021). Recently, (Richtárik et al., 2022) have shown that LAG can be properly studied from compression point of view. They derive optimal convergence guarantees suported by emprirical evaluations. The follow-up work (Doikov et al., 2023) uses similar idea in one node regime. They compute Hessian deterministically once in $1/d$ iterations. Their theoretical analysis also show benefits of exploiting rare updates. Nevertheless, the lazy aggregation in the case of second-order methods remains poorly studied. In our work we make a step towards better understanding of lazy aggregation for Newton-type algorithms.

For more examples of 3PC compressors see section 3 of Richtárik et al. (2022) and the Appendix.

## 4 Newton-3PC: **Newton's Method with 3PC**

In this section we present our first Newton-type method, called Newton-3PC, employing communication compression through 3PC compressors discussed in the previous section. The proposed method is an extension

of FedNL (Safaryan et al., 2022) from contractive compressors to arbitrary 3PC compressors. From this perspective, our Newton-3PC (see Algorithm 1) is much more flexible, offering a wide variety of communication strategies beyond contractive compressors.

### 4.1 General technique for learning the Hessian

The central notion in FedNL is the technique for learning *a priori unknown* Hessian $\nabla^2 f(x^*)$ at the (unique) solution $x^*$ in a communication efficient manner. This is achieved by maintaining and iteratively updating local Hessian estimates $\mathbf{H}_i^k$ of $\nabla^2 f_i(x^*)$ for all devices $i \in [n]$ and the global Hessian estimate $\mathbf{H}^k = \frac{1}{n} \sum_{i=1}^{n} \mathbf{H}_i^k$ of $\nabla^2 f(x^*)$ for the central server. We adopt the same idea of Hessian learning and aim to update local estimates in such a way that $\mathbf{H}_i^k \to \nabla^2 f_i(x^*)$ for all $i \in [n]$, and as a consequence, $\mathbf{H}^k \to \nabla^2 f(x^*)$, throughout the training process. However, in contrast to FedNL, we update local Hessian estimates via generic 3PC mechanism, namely

$$\mathbf{H}_i^{k+1} = \mathcal{C}_{\mathbf{H}_i^k, \nabla^2 f_i(x^k)} \left( \nabla^2 f_i(x^{k+1}) \right),$$

which is a particular instantiation of 3PC compressor $\mathcal{C}_{\mathbf{H}, \mathbf{Y}}(\mathbf{X})$ using previous local Hessian $\mathbf{Y} = \nabla^2 f_i(x^k)$ and previous estimate $\mathbf{H} = \mathbf{H}_i^k$ to compress current local Hessian $\mathbf{X} = \nabla^2 f_i(x^{k+1})$.

---

**Algorithm 1** Newton-3PC (Newton's method with 3PC)

1: **Input:** $x^0 \in \mathbb{R}^d$, $\mathbf{H}_1^0, \ldots, \mathbf{H}_n^0 \in \mathbb{R}^{d \times d}$, $\mathbf{H}^0 := \frac{1}{n} \sum_{i=1}^{n} \mathbf{H}_i^0$, $l^0 = \frac{1}{n} \sum_{i=1}^{n} \|\mathbf{H}_i^0 - \nabla^2 f_i(x^0)\|$.
2: **on** server
3:     *Option 1:* $x^{k+1} = x^k - [\mathbf{H}^k]_\mu^{-1} \nabla f(x^k)$
4:     *Option 2:* $x^{k+1} = x^k - [\mathbf{H}^k + l^k \mathbf{I}]^{-1} \nabla f(x^k)$
5:     Broadcast $x^{k+1}$ to all nodes
6: **for** each device $i = 1, \ldots, n$ in parallel **do**
7:     Get $x^{k+1}$ and compute local gradient $\nabla f_i(x^{k+1})$ and local Hessian $\nabla^2 f_i(x^{k+1})$
8:     Apply 3PC and update local Hessian estimator to $\mathbf{H}_i^{k+1} = \mathcal{C}_{\mathbf{H}_i^k, \nabla^2 f_i(x^k)} \left( \nabla^2 f_i(x^{k+1}) \right)$
9:     Send $\nabla f_i(x^{k+1})$, $\mathbf{H}_i^{k+1}$ to the server          ▷ the latter is sent only if $\mathbf{H}_i^{k+1}$ has been updated
10:    Send $l_i^{k+1} := \|\mathbf{H}_i^{k+1} - \nabla^2 f_i(x^{k+1})\|_{\mathrm{F}}$          ▷ if *Option 2* is used
11: **end for**
12: **on** server
13:    $\mathbf{H}^{k+1} = \frac{1}{n} \sum_{i=1}^{n} \mathbf{H}_i^{k+1}$, $l^{k+1} = \frac{1}{n} \sum_{i=1}^{n} l_i^{k+1}$

---

In the special case, when EF21 scheme $\mathcal{C}_{\mathbf{H}_i^k, \nabla^2 f_i(x^k)} \left( \nabla^2 f_i(x^{k+1}) \right) = \mathbf{H}_i^k + \mathcal{C}(\nabla^2 f_i(x^{k+1}) - \mathbf{H}_i^k)$ is employed as a 3PC mechanism, we recover the Hessian learning technique of FedNL. Our Newton-3PC method also recovers recently proposed *Basis Learn* (BL) (Qian et al., 2022) algorithm if we specialize the 3PC mechanism to *rotation compression* (see Appendix).

### 4.2 Flexible Hessian communication and computation schemes.

The key novelty Newton-3PC brings is the flexibility of options to handle costly local Hessian matrices both in terms of computation and communication.

Due to the adaptive nature of CLAG mechanism (7), Newton-CLAG method *does not send any information* about the local Hessian $\nabla^2 f_i(x^{k+1})$ if it is sufficiently close to previous Hessian estimate $\mathbf{H}_i^k$, namely $\|\nabla^2 f_i(x^{k+1}) - \mathbf{H}_i^k\|_{\mathrm{F}}^2 \leq \zeta \|\nabla^2 f_i(x^{k+1}) - \nabla^2 f_i(x^k)\|_{\mathrm{F}}^2$ with some positive trigger $\zeta > 0$. In other words, the server *reuses* local Hessian estimate $\mathbf{H}_i^k$ while there is no essential discrepancy between locally computed Hessian $\nabla^2 f_i(x^{k+1})$. Once a sufficient change is detected by the device, only the compressed difference $\mathcal{C}(\nabla^2 f_i(x^{k+1}) - \mathbf{H}_i^k)$ is communicated since the server knows $\mathbf{H}_i^k$. By adjusting the trigger $\zeta$, we can control the frequency of Hessian communication in an adaptive manner. Together with adaptive thresholding operator (5) as a contractive compressor, CLAG is a doubly adaptive communication strategy that makes Newton-CLAG highly efficient in terms of communication complexity.

Interestingly enough, we can design such 3PC compressors that can reduce computational costs too. To achieve this, we consider CBAG mechanism (8) which replaces the adaptive switching condition of CLAG by probabilistic switching according to Bernoulli random variable. Due to the probabilistic nature of CBAG mechanism, Newton-CBAG method requires devices to compute local Hessian $\nabla^2 f_i(x^{k+1})$ and communicate compressed difference $\mathcal{C}(\nabla^2 f_i(x^{k+1}) - \mathbf{H}_i^k)$ *only* with probability $p \in (0, 1]$. Otherwise, the whole Hessian computation and communication is *skipped*.

### 4.3 Options for updating the global model

We adopt the same two update rules for the global model as was design in FedNL. If the server knows the strong convexity parameter $\mu > 0$ (see Assumption 4.1), then the global Hessian estimate $\mathbf{H}^k$ is projected onto the set $\left\{ \mathbf{M} \in \mathbb{R}^{d \times d} \colon \mathbf{M}^\top = \mathbf{M}, \ \mu \mathbf{I} \preceq \mathbf{M} \right\}$ to get the projected estimate $[\mathbf{H}^k]_\mu$. Alternatively, all devices additionally compute and send compression errors $l_i^k := \|\mathbf{H}_i^k - \nabla^2 f_i(x^k)\|_{\mathrm{F}}$ (extra float from each device in terms of communication complexity) to the server, which then formulates the regularized estimate $\mathbf{H}^k + l^k \mathbf{I}$ by adding the average error $l^k = \frac{1}{n} \sum_{i=1}^n l_i^k$ to the global Hessian estimate $\mathbf{H}^k$.

### 4.4 Local convergence theory

To derive local[1] theoretical guarantees, we consider the standard assumption that the global objective is strongly convex and local Hessians are Lipschitz continuous.

**Assumption 4.1.** *The average loss $f$ is $\mu$-strongly convex, and all local losses $f_i(x)$ have Lipschitz continuous Hessians with respect to three different matrix norms: spectral, Frobenius and infinity norms, respectively. Formally, we require $\|\nabla^2 f_i(x) - \nabla^2 f_i(y)\| \le L_* \|x - y\|$, $\|\nabla^2 f_i(x) - \nabla^2 f_i(y)\|_{\mathrm{F}} \le L_{\mathrm{F}} \|x - y\|$, $\max_{j,l} |(\nabla^2 f_i(x) - \nabla^2 f_i(y))_{jl}| \le L_\infty \|x - y\|$ to hold for all $i \in [n]$ and $x, y \in \mathbb{R}^d$.*

Define constants $C$ and $D$ depending on which option is used for global model update, namely $C = 2, D = L_*^2$ if *Option 1* is used, and $C = 8, D = (L_* + 2L_{\mathrm{F}})^2$ if *Option 2* is used. We prove three local rates for Newton-3PC: for the squared distance to the solution $\|x^k - x^*\|^2$, and for the Lyapunov function

$$\Phi^k := \mathcal{H}^k + 6(1/A + 3AB)L_{\mathrm{F}}^2 \|x^k - x^*\|^2.$$

where $\mathcal{H}^k := \frac{1}{n} \sum_{i=1}^n \|\mathbf{H}_i^k - \nabla^2 f_i(x^*)\|_{\mathrm{F}}^2$.

We present our theoretical results for local convergence with two stages. For the first stage, we derive convergence rates using specific *locality conditions* for model/Hessian estimation error. In the second stage, we prove that these locality conditions are satisfied for different situations.

**Theorem 4.2.** *Let Assumption 4.1 hold. Assume $\|x^0 - x^*\| \le \frac{\mu}{\sqrt{2D}}$ and $\mathcal{H}^k \le \frac{\mu^2}{4C}$ for all $k \ge 0$. Then,* Newton-3PC *(Algorithm 1) with any 3PC mechanism converges with the following rates:*

$$\|x^k - x^*\|^2 \le \frac{1}{2^k}\|x^0 - x^*\|^2, \tag{9}$$

$$\mathbb{E}\left[\Phi^k\right] \le (1 - \rho)^k \Phi^0, \ \rho = \min\left\{\frac{A}{2}, \frac{1}{3}\right\}, \tag{10}$$

$$\mathbb{E}\left[\frac{\|x^{k+1} - x^*\|^2}{\|x^k - x^*\|^2}\right] \le (1 - \rho)^k \left(C + \frac{AD}{12(1 + 3AB)L_{\mathrm{F}}^2}\right)\frac{\Phi^0}{\mu^2}. \tag{11}$$

Clearly, these rates are independent of the condition number of the problem, and the choice of 3PC can control the parameter $A$. Notice that locality conditions here are upper bounds on the initial model error $\|x^0 - x^*\|$ and the errors $\mathcal{H}^k$ for all $k \ge 0$. It turns out that the latter condition may not be guaranteed in general since it depends on the structure of the 3PC mechanism. Below, we show these locality conditions under some assumptions on 3PC, covering practically all compelling cases. We highlight that the design

---

[1]For discussion on global convergence guarantees we refer to the Appendix I.

of Algorithm 1 allows to keep iterates $x^k$ and Hessian approximations $\mathbf{H}_i^k$ in a neighbourhood of $x^*$ and $\nabla^2 f_i(x^*)$ that is crucially important do derive local convergence guarantees. Moreover, Newton-3PC provably works with biased compression operators that are typically harder to analyze.

**Lemma 4.3** (Deterministic 3PC). *Let the 3PC compressor in* Newton-3PC *be deterministic. Assume the following initial conditions hold: $\|x^0 - x^*\| \leq e_1 := \min\{\frac{A\mu}{\sqrt{8(1+3AB)L_{\mathrm{F}}}}, \frac{\mu}{\sqrt{2D}}\}$ and $\|\mathbf{H}_i^0 - \nabla^2 f_i(x^*)\|_{\mathrm{F}} \leq \frac{\mu}{2\sqrt{C}}$. Then $\|x^k - x^*\| \leq e_1$ and $\|\mathbf{H}_i^k - \nabla^2 f_i(x^*)\|_{\mathrm{F}} \leq \frac{\mu}{2\sqrt{C}}$ for all $k \geq 0$.*

**Lemma 4.4** (CBAG). *Consider CBAG mechanism with only source of randomness from Bernoulli aggregation. Assume $\|x^0 - x^*\| \leq e_2 := \min\{\frac{(1-\sqrt{1-\alpha})\mu}{4\sqrt{C}L_{\mathrm{F}}}, \frac{\mu}{\sqrt{2D}}\}$ and $\|\mathbf{H}_i^0 - \nabla^2 f_i(x^*)\|_{\mathrm{F}} \leq \frac{\mu}{2\sqrt{C}}$. Then $\|x^k - x^*\| \leq e_2$ and $\|\mathbf{H}_i^k - \nabla^2 f_i(x^*)\|_{\mathrm{F}} \leq \frac{\mu}{2\sqrt{C}}$ for all $k \geq 0$.*

## 5 Extension to Bidirectional Compression (Newton-3PC-BC)

In this section, we consider the setup where both directions of communication between devices and the central server are bottleneck. For this setup, we propose Newton-3PC-BC (Algorithm 2) which additionally applies Bernoulli aggregation for gradients (worker to server direction) and another 3PC mechanism for the global model (server to worker direction) employed the master.

Overall, the method integrates three independent communication schemes: workers' 3PC (denoted by $\mathcal{C}^W$) for local Hessian matrices $\nabla^2 f_i(z^{k+1})$, master's 3PC (denoted by $\mathcal{C}^M$) for the global model $x^{k+1}$ and Bernoulli aggregation with probability $p \in (0, 1]$ for local gradients $\nabla f_i(z^{k+1})$. Because of these three mechanisms, the method maintains three sequences of model parameters $\{x^k, w^k, z^k\}_{k \geq 0}$. Parameters $x^k$ and $w^k$ are server's and clients' models respectively while $w^k$ is a copy of $z^k$ when local gradients were last computed. Notice that, Bernoulli aggregation for local gradients is a special case of CBAG (Example 3.9), which allows to skip the computation of local gradients with probability $(1 - p)$. However, this reduction in gradient computation necessitates algorithmic modification in order to guarantee convergence. Specifically, we design gradient estimator $g^{k+1}$ to be the full gradient $\nabla f(z^{k+1})$ if devices compute local gradients (i.e., $\xi = 1$). Otherwise, when gradient computation is skipped (i.e., $\xi = 0$), we estimate the missing gradient using Hessian estimate $\mathbf{H}^{k+1}$ and stale gradient $\nabla f(w^{k+1})$, namely we set $g^{k+1} = [\mathbf{H}^{k+1}]_\mu(z^{k+1} - w^{k+1}) + \nabla f(w^{k+1})$.

Similar to the previous result, we present convergence rates and guarantees for locality separately. Let $A_M(A_W)$, $B_M(B_W)$ be parameters of the master's (workers') 3PC mechanisms. Define constants $C_M := \frac{4}{A_M} + 1 + \frac{5B_M}{2}$, $C_W := \frac{4}{A_W} + 1 + \frac{5B_W}{2}$ and Lyapunov function $\Phi_1^k := \|z^k - x^*\|^2 + C_M \|x^k - x^*\|^2 + \frac{A_M(1-p)}{4p}\|w^k - x^*\|^2$.

**Theorem 5.1.** *Let Assumption 4.1 holds. Assume $\|z^k - x^*\|^2 \leq \frac{A_M \mu^2}{24 C_M L_*^2}$ and $\mathcal{H}^k \leq \frac{A_M \mu^2}{96 C_M}$ for all $k \geq 0$. Then,* Newton-3PC-BC *(Algorithm 2) converges with the following linear rate:*

$$\mathbb{E}[\Phi_1^k] \leq \left(1 - \min\{\tfrac{A_M}{4}, \tfrac{3p}{8}\}\right)^k \Phi_1^0. \tag{12}$$

Note that the above linear rate for $\Phi_1^k$ does not depend on the conditioning of the problem and implies linear rates for all three sequences $\{x^k, w^k, z^k\}$. Next we prove locality conditions used in the theorem for two cases: for non-random 3PC schemes and for schemes that preserve certain convex combination condition. It can be seen easily that random sparsification fits into the second case.

**Lemma 5.2** (Deterministic 3PC). *Let Assumption 4.1 holds. Let $\mathcal{C}^M$ and $\mathcal{C}^W$ be deterministic. Assume $\|x^0 - x^*\|^2 \leq \frac{11 A_M}{24 C_M} e_3^2 := \frac{11 A_M}{24 C_M} \min\{\frac{A_M \mu^2}{24 C_M L_*^2}, \frac{A_W A_M \mu^2}{384 C_W C_M L_{\mathrm{F}}^2}\}$ and $\mathcal{H}^0 \leq \frac{A_M \mu^2}{96 C_M}$. Then $\|x^k - x^*\|^2 \leq \frac{11 A_M}{24 C_M} e_3^2$, $\|z^k - x^*\|^2 \leq e_3^2$ and $\mathcal{H}^k \leq \frac{A_M \mu^2}{96 C_M}$ for all $k \geq 0$*

**Lemma 5.3** (Random sparsification). *Let Assumption 4.1 holds. Assume $(z^k)_j$ is a convex combination of $\{(x^t)_j\}_{t=0}^k$, and $(\mathbf{H}_i^k)_{jl}$ is a convex combination of $\{(\nabla^2 f_i(z^k))_{jl}\}_{t=0}^k$ for all $i \in [n]$, $j, l \in [d]$, and $k \geq 0$. If $\|x^0 - x^*\|^2 \leq e_4^2 := \min\{\frac{\mu^2}{d^2 L_*^2}, \frac{A_M \mu^2}{24 d C_M L_*^2}, \frac{A_M \mu^2}{96 d^3 C_M L_\infty^2}, \frac{\mu^2}{4 d^4 L_\infty^2}\}$, then $\|z^k - x^*\|^2 \leq d e_4^2$ and $\mathcal{H}^k \leq \min\{\frac{A_M \mu^2}{96 C_M}, \frac{\mu^2}{4d}\}$ for all $k \geq 0$.*

---

**Algorithm 2** Newton-3PC-BC (Newton's method with 3PC and Bidirectional Compression)

---

1: **Parameters:** Workers-side 3PC ($\mathcal{C}^W$), Master-side 3PC ($\mathcal{C}^M$), gradient probability $p \in (0, 1]$
2: **Input:** $x^0 = w^0 = z^0 \in \mathbb{R}^d$; $\mathbf{H}_i^0 \in \mathbb{R}^{d \times d}$, and $\mathbf{H}^0 := \frac{1}{n} \sum_{i=1}^n \mathbf{H}_i^0$; $\xi^0 = 1$; $g^0 = \nabla f(z^0)$
3: **on** server
4:     Update the global model to $x^{k+1} = z^k - [\mathbf{H}^k]_\mu^{-1} g^k$
5:     Apply Master-side 3PC and send model estimate $z^{k+1} = \mathcal{C}^M_{z^k, x^k}(x^{k+1})$ to all devices $i \in [n]$
6:     Sample $\xi^{k+1} \sim \text{Bernoulli}(p)$ and send to all $i \in [n]$
7: **for** each device $i = 1, \ldots, n$ in parallel **do**
8:     Get $z^{k+1} = \mathcal{C}^M_{z^k, x^k}(x^{k+1})$ and $\xi^{k+1}$ from the server
9:     **if** $\xi^{k+1} = 1$
10:     $w^{k+1} = z^{k+1}$, compute local gradient $\nabla f_i(z^{k+1})$ and send to the server
11:     **if** $\xi^{k+1} = 0$
12:     $w^{k+1} = w^k$
13:     Apply Worker's 3PC and update local Hessian estimator to $\mathbf{H}_i^{k+1} = \mathcal{C}^W_{\mathbf{H}_i^k, \nabla^2 f_i(z^k)}(\nabla^2 f_i(z^{k+1}))$
14: **end for**
15: **on** server
16:     $\nabla f(z^{k+1}) = \frac{1}{n} \sum_{i=1}^n \nabla f_i(z^{k+1})$, $\mathbf{H}^{k+1} = \frac{1}{n} \sum_{i=1}^n \mathbf{H}_i^k$
17:     **if** $\xi^{k+1} = 1$
18:     $w^{k+1} = z^{k+1}$, $g^{k+1} = \nabla f(z^{k+1})$
19:     **if** $\xi^{k+1} = 0$
20:     $w^{k+1} = w^k$,
21:     $g^{k+1} = [\mathbf{H}^{k+1}]_\mu(z^{k+1} - w^{k+1}) + \nabla f(w^{k+1})$

---

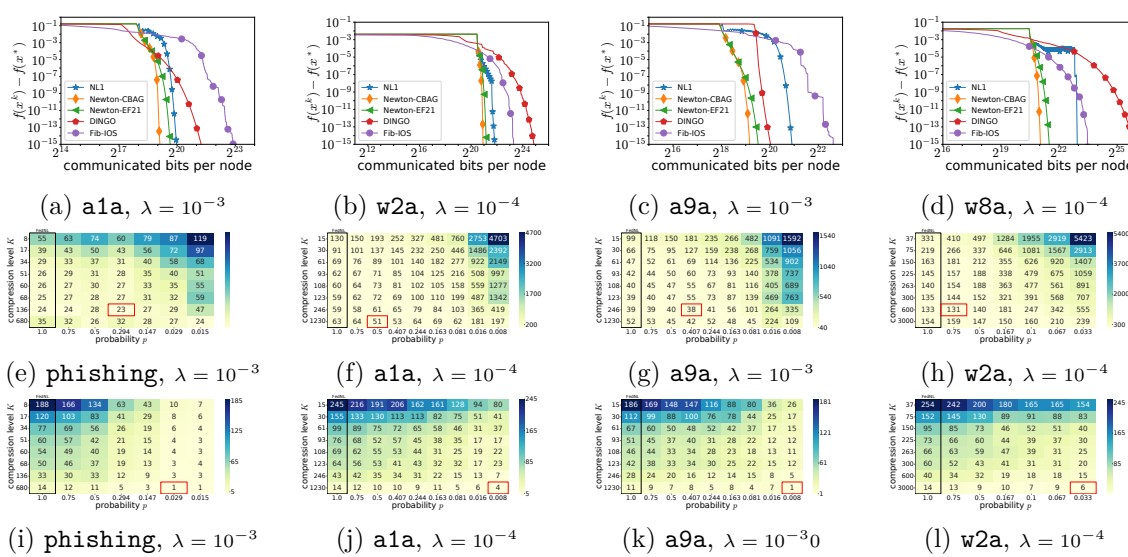

**Figure 1:** Comparison of Newton-CBAG with Top-$d$ compressor and probability $p = 0.75$, Newton-EF21 with Rank-1 compressor, NL1 with Rand-1 compressor, and DINGO (**first row**). The performance of Newton-CBAG with Top-$d$ in terms of communication complexity (**second row**, in Mbytes) and the number of local Hessian computations (**third row**).

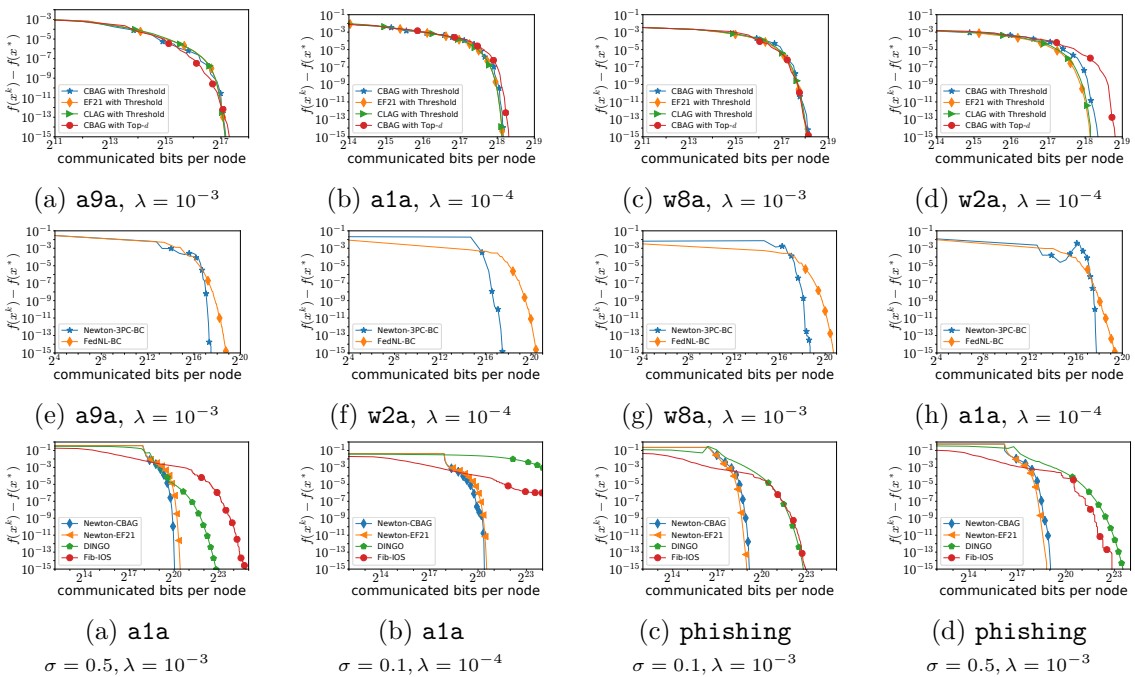

**Figure 2:** Comparison of Newton-CBAG with thresholding and Top-$d$ compressors and Newton-EF21 with thresholding compressor in terms of communication complexity (**first row**). Comparison of Newton-3PC-BC against FedNL-BC in terms of communication complexity (**second row**). The performance of Newton-CBAG combined with Top-$d$ compressor and probability $p = 0.75$, Newton-EF21 with Rank-1 compressor, DINGO, and Fib-ISO in terms of communication complexity on Softmax problem (**third row**).

## 6 Experiments

In this part, we study the empirical performance of Newton-3PC comparing its performance against other second-order methods on logistic regression problems of the form

$$\min_{x \in \mathbb{R}^d} \left\{ f(x) := \frac{1}{n} \sum_{i=1}^{n} f_i(x) + \frac{\lambda}{2} \|x\|^2 \right\}, \tag{13}$$

where $f_i(x) = \frac{1}{m} \sum_{j=1}^{m} \log \left( 1 + \exp(-b_{ij} a_{ij}^\top x) \right)$ and $\{a_{ij}, b_{ij}\}_{j \in [m]}$ are data points belonging to $i$-th client. We use datasets from LibSVM library (Chang & Lin, 2011). Each dataset was shuffled and split into $n$ equal parts. Detailed description of datasets and hyperparameters choice is given in the Appendix.

### 6.1 Comparison between Newton-3PC and other second-order methods

According to Safaryan et al. (2022), FedNL with Rank-1 compressor outperforms other second-order methods in all cases in terms of communication complexity. Thus, we compare in Figure 1 (first row) Newton-CBAG (based on Top-$d$ compressor and probability $p = 0.75$), Newton-EF21 with Rank-1, NL1 with Rand-1, DINGO, and Fib-IOS indicating how many bits are transmitted by each client in both uplink and downlink directions. We clearly see that Newton-CBAG is much more communication efficient than NL1, Fib-IOS and DINGO. Besides, it outperforms FedNL in all cases. On top of that, we achieve improvement not only in communication complexity, but also in computational cost with Newton-CBAG. Indeed, when clients do not send compressed Hessian differences to the server there is no need to compute local Hessians. Consequently, computational costs goes down. We decided not to compare Newton-3PC with first-order methods since FedNL already outperforms them in terms of communication complexity in a variety of experiments in (Safaryan et al., 2022).

## 6.2 Does Bernoulli aggregation bring any advantage?

Next, we investigate the performance of Newton-CBAG based on Top-$K$. We report the results in heatmaps (see Figure 1, second row) where we vary probability $p$ along rows and compression level $K$ along columns. Notice that Newton-CBAG reduces to FedNL when $p = 1$ (left column). We observe that Bernoulli aggregation (BAG) is indeed beneficial since the communication complexity reduces when $p$ becomes smaller than 1 (in case of a1a data set the improvement is significant). We can conclude that BAG leads to better communication complexity of Newton-3PC over FedNL.

On top of that, we claim that Newton-CBAG is also computationally more efficient than FedNL; see Figure 1 (third row) that indicates the number of Hessian computations. We observe that even if communication complexity in two regimes are close to each other, but computationally better the one with smaller $p$. Indeed, in the case when $p < 1$ we do not have to compute local Hessians with probability $1 - p$ that leads to acceleration in terms of computation complexity.

## 6.3 3PC based on adaptive thresholding

Next we test the performance of Newton-3PC using adaptive thresholding operator (5). We compare Newton-EF21 (equivalent to FedNL), Newton-CBAG, and Newton-CLAG with adaptive thresholding against Newton-CBAG with Top-$d$ compressor. We fix the probability $p = 0.5$ for CBAG, the trigger $\zeta = 2$ for CLAG, and thresholding parameter $\lambda = 0.5$. According to the results presented in Figure 2 (first row), adaptive thresholding can be beneficial since it improves the performance of Newton-3PC in some cases. Moreover, it is computationally cheaper than Top-$K$ as we do not sort entries of a matrix as it is for Top-$K$.

## 6.4 Newton-3PC-BC against FedNL-BC

In our next experiment we study bidirectional compression. We compare Newton-3PC-BC against FedNL-BC. For Newton-3PC-BC we fix CBAG with $p = 0.75$ combined with Top-$d$ compressor applied on Hessians, BAG with $p = 0.75$ applied on gradients, and 3PCv4 (Richtárik et al., 2022) combined with (Top-$K_1$, Top-$K_2$) compressors on iterates. For FedNL-BC we use Top-$d$ compressor on Hessians and BAG with $p = 0.75$ on gradients, and Top-$K$ compressor on iterates. We choose different values for $K_1$ and $K_2$ such that it $K_1 + K_2 = K$ always hold. Such choice of parameters allows to make the iteration cost of both methods to be equal. Based on the results, we argue that the superposition of CBAG and 3PCv4 applied on Hessians and iterates respectively is more communication efficient than the combination of EF21 and EF21.

## 6.5 Performance of Newton-3PC on Softmax problem

Finally, we also consider L2 regularized Softmax problem where all $f_i$'s of the form

$$f_i(x) = \sigma \log \left( \sum_{j=1}^{m} \exp \left( \frac{a_{ij}^\top x - b_{ij}}{\sigma} \right) \right).$$

Here $\sigma > 0$ is a smoothing parameter. One can show that this function has both Lipschitz continuous gradient and Lipschitz continuous Hessian. We perform the same data shift as it was done in (Hanzely et al., 2020) (section 8.2). Note that in this case we do not compare Newton-3PC against NL1 as this problem does not belong to the class of *generalized linear models.*

We compare Newton-CBAG combined with Top-$d$ compressor and probability $p = 0.75$, Newton-EF21 with Rank-1 compressor, DINGO (Crane & Roosta, 2019), and Fib-IOS (Fabbro et al., 2022). As we can see in Figure 2 (third row), Newton-CBAG and Newton-EF21 demonstrate almost equivalent performance: in some cases slightly better the first one (a1a dataset), in some cases — the second (phishing dataset). Furthermore, DINGO and Fib-IOS are significantly slower than Newton-3PC methods in terms of communication complexity.

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

## A  Appendix

## B  Limitations

Here we discuss key limitations of our work and areas that are not explored in this paper.

- Developed theoretical claims are for strongly convex loss functions. The globalization mechanism with cubic regularization can be anaylzed for convex functions as well, but we do not consider non-convex objectives in this work.

- Our methods are analyzed in the regime when the exact local gradients and exact local Hessians of local loss functions are computed for all participating devices. We do not consider stochastic gradient or stochastic Hessian oracles of local loss functions in our analyses. However, when we use sketch-and-project operator we rely on Hessian-vector products which does not require full Hessian computations.

## C  Detailed Literature Review of Second-Order Methods

In this section we provide more detailed literature review of second-order methods. The comparison is made based on the most relevant prior works in the literature highlighting main differences over our work. The comparison is performed based on criterias including generality of the considered problem structure, assumptions made on the (local) loss functions, communication complexity per iteration, theoretical convergence guarantees and other aspects of the method.

- GIANT (Wang et al., 2018) and NL (Islamov et al., 2021) are not designed to handle a general finite sum problem. In contrast to our work, they only work with Generalized Linear models.

- Communication costs per iteration of DAN (Zhang et al., 2020a) and Quantized Newton (Alimisis et al., 2021) are significantly high which make them impractical.

- NL (Islamov et al., 2021) directly reveals local data in each iteration which breaks privacy preserving guarantees.

- The drawback of GIANT, DINGO is in their convergence rates which depend on the condition number of the problem. In some cases theoretical convergence guarantees are even worse than those of first-order methods. DANE (Shamir et al., 2014) and AIDE (Reddi et al., 2016) suffer from the same problem because those methods are first-order methods.

- The first drawback of FLECS (Agafonov et al., 2022) is that SVD decomposition is needed in each step to perform a truncation which means that the computation cost of those methods can not be reduced. Besides, there is no good local theory for that method; the only convergence guarantee is derived under the assumption that the iterates remain close to the optimum. Next, convergence of FLECS and FLECS-SGD depend on the product of the condition number and truncation parameters. For example, using the same truncation parameters as in their experiments, the convergence guarantees are of the order $10^{22}\kappa$, where $\kappa$ is the condition number. Finally, they need backtracking line search or other learning techniques with additional parameters to perform one step of the methods.

- Fib-IOS (Fabbro et al., 2022) introduces Newton-type method based on SVD decomposition which means that similarly to FLECS the computation cost can not be reduced. Besides, this approach is also restricted to rank-type compression only, and consequently does not support popular compression techniques such as Top-$K$ or Rand-$K$. On top of that, Fib-IOS always requires backtracking line search technique to find appropriate stepsize.

- GIANT and DANE work well only in homogeneous setting while in practice the problem could be significantly heterogeneous. In our work we do not make any assumptions on heterogeneity of the problem.

- While convergence guarantees of FedNL (Safaryan et al., 2022) and Newton-3PC are the same (fast local linear/superlinear rates independent of the condition number) there are several points that make our work superior: (*i*) FedNL can be seen as a special case of Newton-3PC; (*ii*) we provide much wider compression mechanisms going beyond those proposed in (Safaryan et al., 2022); (*iii*) we propose two ways how to reduce computation costs (lazy aggregation and sketching) of Newton-3PC.

## D   More on Sketch-and-Project Mechanism

Sketch-and-project operator has been widely studied as a technique to solve linear systems (Richtárik & Takac, 2017; Gower & Richtárik, 2015), an application to first-order methods (Hanzely et al., 2018). Besides, Gower & Richtárik (2017) showed that various Quasi-Newton updates can be seen as a special case of sketch-and-project mechanism. Let us now describe it in more details. For this reason, we introduce an arbitrary twice differentiable function $\varphi : \mathbb{R}^d \to \mathbb{R}$. Our desire is to compute an approximation of its Hessian $\nabla^2 \varphi$ at point $x$. Let $\mathbf{X}_0$ be the first approximation of $\nabla^2 \varphi(x)$. Then we define a sequence $\{\mathbf{X}_k\}$ that approximates $\nabla^2 \varphi(x)$ better and better as $k$ goes to infinity solving the following optimization problem

$$\mathbf{X}_{k+1} := \mathrm{argmin}_{\mathbf{X} \in \mathbb{R}^{d \times d}} \frac{1}{2} \|\mathbf{X} - \mathbf{X}_k\|_{\mathrm{F}}^2 \quad \text{s.t.} \quad \mathbf{S}^\top \nabla^2 \varphi(x) = \mathbf{S}^\top \mathbf{X}, \tag{14}$$

where $\mathbf{S} \in \mathbb{R}^{d \times \tau}$ is a random matrix drawn in i.i.d. fashion from a fixed distribution $\mathcal{D}$. To solve this problem we define a function $\mathrm{vec}(\mathbf{A}) = (\mathbf{A}_{11}, \dots, \mathbf{A}_{d1}, \mathbf{A}_{12}, \dots, \mathbf{A}_{d2}, \dots, \mathbf{A}_{1d}, \dots, \mathbf{A}_{dd})^\top$. Moreover, we need to define an extended sketch matrix $\tilde{\mathbf{S}}$ of the form

$$\tilde{\mathbf{S}} := \begin{pmatrix} \mathbf{S} & \mathbf{0} & \dots & \mathbf{0} \\ \mathbf{0} & \mathbf{S} & \dots & \mathbf{0} \\ \vdots & \vdots & \ddots & \vdots \\ \mathbf{0} & \mathbf{0} & \dots & \mathbf{S} \end{pmatrix} \in \mathbb{R}^{d^2 \times d\tau}. \tag{15}$$

Using (15), we can reformulate (14) as follows

$$\mathrm{vec}(\mathbf{X}_{k+1}) = \mathrm{argmin}_{\mathbf{X}} \|\mathrm{vec}(\mathbf{X}) - \mathrm{vec}(\mathbf{X}_k)\|^2, \quad \tilde{\mathbf{S}}^\top \mathrm{vec}(\mathbf{X}) = \tilde{\mathbf{S}}^\top \mathrm{vec}(\nabla^2 \varphi(x)). \tag{16}$$

The latter has an explicit solution (Hanzely et al., 2018) of the following form

$$\mathrm{vec}(\mathbf{X}_{k+1}) = \mathrm{vec}(\mathbf{X}_k) + \tilde{\mathbf{Z}}(\mathrm{vec}(\nabla^2 \varphi(x)) - \mathrm{vec}(\mathbf{X}_k)), \tag{17}$$

where $\tilde{\mathbf{Z}} := \tilde{\mathbf{S}}(\tilde{\mathbf{S}}^\top \tilde{\mathbf{S}})^\dagger \tilde{\mathbf{S}}^\top$. It is easy to show that $\tilde{\mathbf{Z}}$ can be rewritten as follows

$$\tilde{\mathbf{Z}} = \begin{pmatrix} \mathbf{Z} & \mathbf{0} & \dots & \mathbf{0} \\ \mathbf{0} & \mathbf{Z} & \dots & \mathbf{0} \\ \vdots & \vdots & \ddots & \vdots \\ \mathbf{0} & \mathbf{0} & \dots & \mathbf{Z} \end{pmatrix}, \tag{18}$$

where $\mathbf{Z} := \mathbf{S}(\mathbf{S}^\top \mathbf{S})^\dagger \mathbf{S}^\top$ is a projection matrix onto the range of a sketch $\mathbf{S}$. Since it is not clear how to compute the process (17), we rewrite it explicitly as

$$\mathbf{X}_{k+1} = \mathbf{X}_k + \mathbf{S}(\mathbf{S}^\top \mathbf{S})^\dagger (\nabla^2 \varphi(x) - \mathbf{X}_k).$$

The way in which the above formula is written resembles the update from (Safaryan et al., 2022).

## E   Deferred Proofs from Section 3 and New 3PC Compressors

### E.1   Proof of Lemma 3.4: Adaptive Thresholding

Basically, we show two upper bounds for the error and combine them to get the expression for $\alpha$. From the definition (5), we get

$$\|\mathcal{C}(\mathbf{X}) - \mathbf{X}\|_{\mathrm{F}}^2 = \sum_{j,l:|\mathbf{X}_{jl}| < \lambda \|\mathbf{X}\|_\infty} \mathbf{X}_{jl}^2 \le d^2 \lambda^2 \|\mathbf{X}\|_\infty^2 \le d^2 \lambda^2 \|\mathbf{X}\|_{\mathrm{F}}^2.$$

The second inequality is derived from the observation that at least on entry, the top one in magnitude, is selected always. Since the top entry is missing in the sum below, we imply that the average without the top one is smaller than the overall average.

$$\|\mathcal{C}(\mathbf{X}) - \mathbf{X}\|_{\mathrm{F}}^2 = \sum_{j,l:|\mathbf{X}_{jl}|<\lambda\|\mathbf{X}\|_{\infty}} \mathbf{X}_{jl}^2 \leq \frac{d^2-1}{d^2} \sum_{j,l=1}^{d} \mathbf{X}_{jl}^2 \leq \left(1 - \frac{1}{d^2}\right)\|\mathbf{X}\|_{\mathrm{F}}^2.$$

### E.2 Proof of Lemma 3.6: Sketch-and-Project

Note that since $\mathbf{Z}$ is a projection matrix, then it is symmetric and satisfies

$$\begin{aligned} \mathbf{ZZ} &= \mathbf{S}(\mathbf{S}^\top \mathbf{S})^\dagger \mathbf{S}^\top \mathbf{S}(\mathbf{S}^\top \mathbf{S})^\dagger \mathbf{S}^\top \\ &= \mathbf{S}(\mathbf{S}^\top \mathbf{S})^\dagger \mathbf{S}^\top = \mathbf{Z}, \end{aligned}$$

As a consequence, its eigenvalues are between 0 and 1. Assuming that $\mathbf{X}$ is symmetric we derive

$$\begin{aligned} \mathbb{E}\left[\|\mathbf{ZX} - \mathbf{X}\|_{\mathrm{F}}^2\right] &= \mathbb{E}\left[\|\mathbf{ZX}\|_{\mathrm{F}}^2 - 2\langle \mathbf{ZX}, \mathbf{X}\rangle\right] + \|\mathbf{X}\|_{\mathrm{F}}^2 \\ &= \mathbb{E}\left[\mathrm{Tr}(\mathbf{X}^\top \mathbf{Z}^\top \mathbf{ZX})\right] - 2\langle \mathbb{E}\left[\mathbf{Z}\right]\mathbf{X}, \mathbf{X}\rangle + \|\mathbf{X}\|_{\mathrm{F}}^2 \\ &= \mathbb{E}\left[\langle \mathbf{X}, \mathbf{ZX}\rangle\right] - 2\langle \mathbb{E}\left[\mathbf{Z}\right]\mathbf{X}, \mathbf{X}\rangle + \|\mathbf{X}\|_{\mathrm{F}}^2 \\ &= \|\mathbf{X}\|_{\mathrm{F}}^2 - \mathrm{Tr}(\mathbf{X}^\top \mathbb{E}\left[\mathbf{Z}\right]\mathbf{X}) \\ &\leq (1 - \lambda_{\min}^+(\mathbf{Z}))\|\mathbf{X}\|_{\mathrm{F}}^2. \end{aligned}$$

Note that we can force $\mathbf{X}$ to be symmetric in the same way as it was done in (Qian et al., 2022) by using symmetrization operator $[\mathbf{X}]_s = \frac{1}{2}(\mathbf{X} + \mathbf{X}^\top)$ which does not change the theory.

### E.3 Proof of Lemma 3.10: Compressed Bernoulli AGgregation (CBAG)

As it was mentioned CBAG has two independent sources of randomness: Bernoulli aggregation and possible random contractive compression. To show that CBAG is a 3PC mechanism, we consider these randomness one by one and upper bound the error as follows:

$$\begin{aligned} \mathbb{E}\left[\|\mathcal{C}_{\mathbf{H},\mathbf{Y}}(\mathbf{X}) - \mathbf{X}\|^2\right] &= (1-p)\|\mathbf{H} - \mathbf{X}\|^2 + p\mathbb{E}\left[\|\mathcal{C}(\mathbf{X} - \mathbf{H}) - (\mathbf{X} - \mathbf{H})\|^2\right] \\ &\leq (1-p)\|\mathbf{X} - \mathbf{H}\|^2 + p(1-\alpha)\|\mathbf{X} - \mathbf{H}\|^2 \\ &= (1-p\alpha)\|\mathbf{X} - \mathbf{H}\|^2 \\ &\leq (1-p\alpha)(1+s)\|\mathbf{H} - \mathbf{Y}\|^2 + (1-p\alpha)(1+1/s)\|\mathbf{X} - \mathbf{Y}\|^2. \end{aligned}$$

### E.4 New 3PC: Adaptive Top-$K$

Assume that in our framework we are restricted by the number of floats we can send from clients to the server. For example, each client is able to broadcast $d_0 \leq d^2$ floats to the server. Besides, we want to use Top-$K$ compression operator with adaptive $K$, but due to the aforementioned restrictions we should control how $K$ evolves. Let $K_{\mathbf{H},\mathbf{Y}}$ be such that

$$K_{\mathbf{H},\mathbf{Y}} = \min\left\{\left\lceil\frac{\|\mathbf{Y} - \mathbf{H}\|_{\mathrm{F}}^2}{\|\mathbf{X} - \mathbf{H}\|_{\mathrm{F}}^2}d^2\right\rceil, d_0\right\}.$$

We introduce the following compression operator

$$\mathcal{C}_{\mathbf{H},\mathbf{Y}}(\mathbf{X}) := \mathbf{H} + \text{Top-}K_{\mathbf{H},\mathbf{Y}}(\mathbf{X} - \mathbf{H}). \tag{19}$$

The next lemma shows that the described compressor satisfy (3).

**Lemma E.1.** *The compressor* $\mathcal{C}_{\mathbf{Y},\mathbf{H}}$ (19) *satisfy* (3) *with*

$$A = \frac{d_0}{2d^2}, \quad B = \max\left\{\left(1 - \frac{d_0}{d^2}\right)\left(\frac{2d^2}{d_0} - 1\right), 3\right\}.$$

*Proof.* Recall that if $\mathcal{C}$ is a Top-$K$ compressor, then for all $\mathbf{X} \in \mathbb{R}^{d \times d}$

$$\|\mathcal{C}(\mathbf{X}) - \mathbf{X}\|_{\mathrm{F}}^2 \leq \left(1 - \frac{K}{d^2}\right) \|\mathbf{X}\|_{\mathrm{F}}^2,$$

Using this property we get in the case when $K_{\mathbf{Y},\mathbf{H}} = d_0$

$$
\begin{aligned}
\|\mathcal{C}_{\mathbf{H},\mathbf{Y}}(\mathbf{X}) - \mathbf{X}\|_{\mathrm{F}}^2 &= \|\mathbf{H} + \text{Top-}K_{\mathbf{H},\mathbf{Y}}(\mathbf{X} - \mathbf{H}) - \mathbf{X}\|_{\mathrm{F}}^2 \\
&\leq \left(1 - \frac{d_0}{d^2}\right) \|\mathbf{H} - \mathbf{X}\|_{\mathrm{F}}^2 \\
&\leq \left(1 - \frac{d_0}{2d^2}\right) \|\mathbf{H} - \mathbf{Y}\|_{\mathrm{F}}^2 + \left(1 - \frac{d_0}{d^2}\right) \frac{2d^2 - d_0}{d_0} \|\mathbf{Y} - \mathbf{X}\|_{\mathrm{F}}^2.
\end{aligned}
$$

If $K_{\mathbf{H},\mathbf{Y}} = \left\lceil \frac{\|\mathbf{Y}-\mathbf{H}\|_{\mathrm{F}}^2}{\|\mathbf{X}-\mathbf{H}\|_{\mathrm{F}}^2} d^2 \right\rceil$, then $-K_{\mathbf{H},\mathbf{Y}} \leq -\frac{\|\mathbf{Y}-\mathbf{H}\|_{\mathrm{F}}^2}{\|\mathbf{X}-\mathbf{H}\|_{\mathrm{F}}^2} d^2$, and we have

$$
\begin{aligned}
\|\mathcal{C}_{\mathbf{H},\mathbf{Y}}(\mathbf{X}) - \mathbf{X}\|_{\mathrm{F}}^2 &= \|\mathbf{H} + \text{Top-}K_{\mathbf{H},\mathbf{Y}}(\mathbf{X} - \mathbf{H}) - \mathbf{X}\|_{\mathrm{F}}^2 \\
&\leq \left(1 - \frac{K_{\mathbf{H},\mathbf{Y}}}{d^2}\right) \|\mathbf{H} - \mathbf{X}\|_{\mathrm{F}}^2 \\
&\leq \left(1 - \frac{\|\mathbf{Y} - \mathbf{H}\|_{\mathrm{F}}^2}{\|\mathbf{X} - \mathbf{H}\|_{\mathrm{F}}^2}\right) \|\mathbf{H} - \mathbf{X}\|_{\mathrm{F}}^2 \\
&= \|\mathbf{H} - \mathbf{X}\|_{\mathrm{F}}^2 - \|\mathbf{Y} - \mathbf{H}\|_{\mathrm{F}}^2 \\
&\leq \frac{3}{2} \|\mathbf{H} - \mathbf{Y}\|_{\mathrm{F}}^2 + 3 \|\mathbf{Y} - \mathbf{X}\|_{\mathrm{F}}^2 - \|\mathbf{Y} - \mathbf{H}\|_{\mathrm{F}}^2 \\
&= \frac{1}{2} \|\mathbf{Y} - \mathbf{H}\|_{\mathrm{F}}^2 + 3 \|\mathbf{Y} - \mathbf{X}\|_{\mathrm{F}}^2,
\end{aligned}
$$

where in the last inequality we use Young's inequality. Since we always have $\frac{d_0}{2d^2}$ (because $d_0 \leq d^2$), then $A = \frac{d_0}{2d^2}$. $\qquad\square$

### E.5  New 3PC: Rotation Compression

Qian et al. (2022) proposed a novel idea to change the basis in the space of matrices that allows to apply more aggresive compression mechanism. Following Section 2.3 from (Qian et al., 2022) one can show that for Generalized Linear Models local Hessians can be represented as $\nabla^2 f_i(x) = \mathbf{Q}_i \Lambda_i(x) \mathbf{Q}_i^\top$, where $\mathbf{Q}_i$ is properly designed basis matrix. This means that $\mathbf{Q}_i$ is orthogonal matrix. Their idea is based on the fact that $\Lambda_i(x)$ is potentially sparser matrix than $\nabla^2 f_i(x)$, and applying compression on $\Lambda_i(x)$ could require smaller compression level to obtain the same results than applying compression on dense standard representation $\nabla^2 f_i(x)$. We introduce the following compression based on this idea. Let $\mathcal{C}$ be an arbitrary contractive compressor with parameter $\alpha$, and $\mathbf{Q}$ be an orthogonal matrix, then our new compressor is defined as follows

$$\mathcal{C}_{\mathbf{H},\mathbf{Y}}(\mathbf{X}) := \mathbf{H} + \mathbf{Q}\mathcal{C}\left(\mathbf{Q}^\top(\mathbf{X} - \mathbf{H})\mathbf{Q}\right)\mathbf{Q}^\top. \tag{20}$$

Now we prove that this compressor satisfy (3).

**Lemma E.2.** *The compressor $\mathcal{C}_{\mathbf{H},\mathbf{Q}}$ (20) based on a contractive compressor $\mathcal{C}$ with parameter $\alpha \in (0,1]$ satisfy (3) with $A = \alpha/2$ and $B = (1 - \alpha)\left((2-\alpha)/\alpha\right)$.*

*Proof.* From the definition of contractive compressor

$$\mathbb{E}\left[\|\mathcal{C}(\mathbf{X}) - \mathbf{X}\|_{\mathrm{F}}^2\right] \leq (1 - \alpha) \|\mathbf{X}\|_{\mathrm{F}}^2.$$

$\qquad\square$

Thus, we get

$$
\begin{aligned}
\mathbb{E}\left[\|\mathcal{C}_{\mathbf{H},\mathbf{Y}}(\mathbf{X}) - \mathbf{X}\|_{\mathrm{F}}^2\right] &= \mathbb{E}\left[\left\|\mathbf{Q}\mathcal{C}\left(\mathbf{Q}^\top(\mathbf{X} - \mathbf{H})\mathbf{Q}\right)\mathbf{Q}^\top - (\mathbf{X} - \mathbf{H})\right\|_{\mathrm{F}}^2\right] \\
&= \mathbb{E}\left[\left\|\mathbf{Q}\mathcal{C}\left(\mathbf{Q}^\top(\mathbf{X} - \mathbf{H})\mathbf{Q}\right)\mathbf{Q}^\top - \mathbf{Q}\mathbf{Q}^\top(\mathbf{X} - \mathbf{H})\mathbf{Q}\mathbf{Q}^\top\right\|_{\mathrm{F}}^2\right] \\
&= \mathbb{E}\left[\left\|\mathcal{C}\left(\mathbf{Q}^\top(\mathbf{X} - \mathbf{H})\mathbf{Q}\right) - \mathbf{Q}^\top(\mathbf{X} - \mathbf{H})\mathbf{Q}\right\|_{\mathrm{F}}^2\right] \\
&\leq (1-\alpha)\left\|\mathbf{Q}^\top(\mathbf{X} - \mathbf{H})\mathbf{Q}\right\|_{\mathrm{F}}^2 \\
&= (1-\alpha)\left\|\mathbf{X} - \mathbf{H}\right\|_{\mathrm{F}}^2 \\
&\leq (1-\alpha)(1+\beta)\left\|\mathbf{Y} - \mathbf{H}\right\|_{\mathrm{F}}^2 + (1-\alpha)(1+\beta^{-1})\left\|\mathbf{Y} - \mathbf{X}\right\|_{\mathrm{F}}^2,
\end{aligned}
$$

where we use the fact that an orthogonal matrix doesn't change a norm. Let $\beta = \frac{\alpha}{2(1-\alpha)}$, then

$$
\mathbb{E}\left[\|\mathcal{C}_{\mathbf{H},\mathbf{Y}}(\mathbf{X}) - \mathbf{X}\|_{\mathrm{F}}^2\right] \leq \left(1 - \frac{\alpha}{2}\right)\|\mathbf{Y} - \mathbf{H}\|_{\mathrm{F}}^2 + (1-\alpha)\left(\frac{2-\alpha}{\alpha}\right)\|\mathbf{Y} - \mathbf{X}\|_{\mathrm{F}}^2. \tag{21}
$$

## F Deferred Proofs from Section 4 (Newton-3PC)

### F.1 Auxiliary lemma

Denote by $\mathbb{E}_{k+1}[\cdot]$ the conditional expectation given $(k+1)^{th}$ iterate $x^{k+1}$. We first develop a lemma to handle the mismatch $\mathbb{E}_k\|\mathbf{H}_i^{k+1} - \nabla^2 f_i(x^*)\|_{\mathrm{F}}^2$ of the estimate $\mathbf{H}_i^{k+1}$ defined via 3PC compressor.

**Lemma F.1.** *Assume that* $\left\|x^{k+1} - x^*\right\|^2 \leq \frac{1}{2}\left\|x^k - x^*\right\|^2$ *for all* $k \geq 0$*. Then*

$$
\mathbb{E}_{k+1}\left[\|\mathbf{H}_i^{k+1} - \nabla^2 f_i(x^*)\|_{\mathrm{F}}^2\right] \leq \left(1 - \frac{A}{2}\right)\|\mathbf{H}_i^k - \nabla^2 f_i(x^*)\|_{\mathrm{F}}^2 + \left(\frac{1}{A} + 3B\right)L_{\mathrm{F}}^2\|x^k - x^*\|_{\mathrm{F}}^2
$$

*Proof.* Using the defining inequality of 3PC compressor and the assumption of the error in terms of iterates, we expand the approximation error of the estimate $\mathbf{H}_i^{k+1}$ as follows:

$$
\begin{aligned}
&\mathbb{E}_{k+1}\left[\|\mathbf{H}_i^{k+1} - \nabla^2 f_i(x^*)\|_{\mathrm{F}}^2\right] \\
=\ & \mathbb{E}_{k+1}\left[\|\mathcal{C}_{\mathbf{H}_i^k, \nabla^2 f_i(x^k)}\left(\nabla^2 f_i(x^{k+1})\right) - \nabla^2 f_i(x^*)\|_{\mathrm{F}}^2\right] \\
\leq\ & (1+\beta)\mathbb{E}_{k+1}\left[\|\mathcal{C}_{\mathbf{H}_i^k, \nabla^2 f_i(x^k)}\left(\nabla^2 f_i(x^{k+1})\right) - \nabla^2 f_i(x^{k+1})\|_{\mathrm{F}}^2\right] + (1+1/\beta)\|\nabla^2 f_i(x^{k+1}) - \nabla^2 f_i(x^*)\|_{\mathrm{F}}^2 \\
\leq\ & (1+\beta)(1-A)\|\mathbf{H}_i^k - \nabla^2 f_i(x^k)\|_{\mathrm{F}}^2 + B\|\nabla^2 f_i(x^{k+1}) - \nabla^2 f_i(x^*)\|_{\mathrm{F}}^2 + (1+1/\beta)\|\nabla^2 f_i(x^{k+1}) - \nabla^2 f_i(x^*)\|_{\mathrm{F}}^2 \\
\leq\ & (1+\beta)(1-A)\|\mathbf{H}_i^k - \nabla^2 f_i(x^k)\|_{\mathrm{F}}^2 \\
& +2B\|\nabla^2 f_i(x^k) - \nabla^2 f_i(x^*)\|_{\mathrm{F}}^2 + (1+1/\beta + 2B)\|\nabla^2 f_i(x^{k+1}) - \nabla^2 f_i(x^*)\|_{\mathrm{F}}^2 \\
\leq\ & (1+\beta)(1-A)\|\mathbf{H}_i^k - \nabla^2 f_i(x^k)\|_{\mathrm{F}}^2 \\
& +2BL_{\mathrm{F}}^2\|x^k - x^*\|_{\mathrm{F}}^2 + (1+1/\beta + 2B)L_{\mathrm{F}}^2\|x^{k+1} - x^*\|_{\mathrm{F}}^2 \\
\leq\ & (1+\beta)(1-A)\|\mathbf{H}_i^k - \nabla^2 f_i(x^k)\|_{\mathrm{F}}^2 + \left(\frac{\beta+1}{2\beta} + 3B\right)L_{\mathrm{F}}^2\|x^k - x^*\|_{\mathrm{F}}^2.
\end{aligned}
$$

where we use Young's inequality for some $\beta > 0$. By choosing $\beta = \frac{A}{2(1-A)}$ when $0 < A < 1$, we get

$$
\mathbb{E}_{k+1}\left[\|\mathbf{H}_i^{k+1} - \nabla^2 f_i(x^*)\|_{\mathrm{F}}^2\right] \leq \left(1 - \frac{A}{2}\right)\|\mathbf{H}_i^k - \nabla^2 f_i(x^*)\|_{\mathrm{F}}^2 + \left(\frac{1}{A} + 3B - \frac{1}{2}\right)L_{\mathrm{F}}^2\|x^k - x^*\|_{\mathrm{F}}^2
$$

When $A = 1$, we can choose $\beta = 1$ and have

$$
\mathbb{E}_{k+1}\left[\|\mathbf{H}_i^{k+1} - \nabla^2 f_i(x^*)\|_{\mathrm{F}}^2\right] \leq (3B+1)L_{\mathrm{F}}^2\|x^k - x^*\|_{\mathrm{F}}^2.
$$

Thus, for all $0 < A \leq 1$ we get the desired bound. $\qquad\square$

### F.2 Proof of Theorem 4.2

The proof follows the same steps as for FedNL until the appearance of 3PC compressor. We derive recurrence relation for $\|x^k - x^*\|^2$ covering both options of updating the global model. If *Option 1.* is used in FedNL, then

$$
\begin{aligned}
\|x^{k+1} - x^*\|^2 &= \left\| x^k - x^* - \left[\mathbf{H}_\mu^k\right]^{-1} \nabla f(x^k) \right\|^2 \\
&\leq \left\| \left[\mathbf{H}_\mu^k\right]^{-1} \right\|^2 \left\| \mathbf{H}_\mu^k(x^k - x^*) - \nabla f(x^k)) \right\|^2 \\
&\leq \frac{2}{\mu^2} \left( \left\| (\mathbf{H}_\mu^k - \nabla^2 f(x^*))(x^k - x^*) \right\|^2 + \left\| \nabla^2 f(x^*)(x^k - x^*) - \nabla f(x^k) + \nabla f(x^*) \right\|^2 \right) \\
&= \frac{2}{\mu^2} \left( \left\| (\mathbf{H}_\mu^k - \nabla^2 f(x^*))(x^k - x^*) \right\|^2 + \left\| \nabla f(x^k) - \nabla f(x^*) - \nabla^2 f(x^*)(x^k - x^*) \right\|^2 \right) \\
&\leq \frac{2}{\mu^2} \left( \left\| \mathbf{H}_\mu^k - \nabla^2 f(x^*) \right\|^2 \|x^k - x^*\|^2 + \frac{L_*^2}{4} \|x^k - x^*\|^4 \right) \\
&= \frac{2}{\mu^2} \|x^k - x^*\|^2 \left( \left\| \mathbf{H}_\mu^k - \nabla^2 f(x^*) \right\|^2 + \frac{L_*^2}{4} \|x^k - x^*\|^2 \right) \\
&\leq \frac{2}{\mu^2} \|x^k - x^*\|^2 \left( \left\| \mathbf{H}^k - \nabla^2 f(x^*) \right\|^2 + \frac{L_*^2}{4} \|x^k - x^*\|^2 \right) \\
&\leq \frac{2}{\mu^2} \|x^k - x^*\|^2 \left( \left\| \mathbf{H}^k - \nabla^2 f(x^*) \right\|_{\mathrm{F}}^2 + \frac{L_*^2}{4} \|x^k - x^*\|^2 \right),
\end{aligned}
$$

where we use $\mathbf{H}_\mu^k \succeq \mu\mathbf{I}$ in the second inequality, and $\nabla^2 f(x^*) \succeq \mu\mathbf{I}$ in the fourth inequality. From the convexity of $\|\cdot\|_{\mathrm{F}}^2$, we have

$$
\|\mathbf{H}^k - \nabla^2 f(x^*)\|_{\mathrm{F}}^2 = \left\| \frac{1}{n} \sum_{i=1}^n \left(\mathbf{H}_i^k - \nabla^2 f_i(x^*)\right) \right\|_{\mathrm{F}}^2 \leq \frac{1}{n} \sum_{i=1}^n \|\mathbf{H}_i^k - \nabla^2 f_i(x^*)\|_{\mathrm{F}}^2 = \mathcal{H}^k.
$$

Thus,

$$
\|x^{k+1} - x^*\|^2 \leq \frac{2}{\mu^2} \|x^k - x^*\|^2 \mathcal{H}^k + \frac{L_*^2}{2\mu^2} \|x^k - x^*\|^4. \tag{22}
$$

If *Option 2.* is used in FedNL, then as $\mathbf{H}^k + l^k\mathbf{I} \succeq \nabla^2 f(x^k) \succeq \mu\mathbf{I}$ and $\nabla f(x^*) = 0$, we have

$$
\begin{aligned}
\|x^{k+1} - x^*\| &= \|x^k - x^* - [\mathbf{H}^k + l^k\mathbf{I}]^{-1}\nabla f(x^k)\| \\
&\leq \|[\mathbf{H}^k + l^k\mathbf{I}]^{-1}\| \cdot \|(\mathbf{H}^k + l^k\mathbf{I})(x^k - x^*) - \nabla f(x^k) + \nabla f(x^*)\| \\
&\leq \frac{1}{\mu}\|(\mathbf{H}^k + l^k\mathbf{I} - \nabla^2 f(x^*))(x^k - x^*)\| + \frac{1}{\mu}\|\nabla f(x^k) - \nabla f(x^*) - \nabla^2 f(x^*)(x^k - x^*)\| \\
&\leq \frac{1}{\mu}\|\mathbf{H}^k + l^k\mathbf{I} - \nabla^2 f(x^*)\|\|x^k - x^*\| + \frac{L_*}{2\mu}\|x^k - x^*\|^2 \\
&\leq \frac{1}{n\mu} \sum_{i=1}^n \|\mathbf{H}_i^k + l_i^k\mathbf{I} - \nabla^2 f_i(x^*)\|\|x^k - x^*\| + \frac{L_*}{2\mu}\|x^k - x^*\|^2 \\
&\leq \frac{1}{n\mu} \sum_{i=1}^n (\|\mathbf{H}_i^k - \nabla^2 f_i(x^*)\| + l_i^k)\|x^k - x^*\| + \frac{L_*}{2\mu}\|x^k - x^*\|^2.
\end{aligned}
$$

From the definition of $l_i^k$, we have

$$
l_i^k = \|\mathbf{H}_i^k - \nabla^2 f_i(x^k)\|_{\mathrm{F}} \leq \|\mathbf{H}_i^k - \nabla^2 f_i(x^*)\|_{\mathrm{F}} + L_{\mathrm{F}}\|x^k - x^*\|.
$$

Thus,

$$\|x^{k+1} - x^*\| \leq \frac{2}{n\mu} \sum_{i=1}^{n} \|\mathbf{H}_i^k - \nabla^2 f_i(x^*)\|_{\mathrm{F}} \|x^k - x^*\| + \frac{L_* + 2L_{\mathrm{F}}}{2\mu} \|x^k - x^*\|^2.$$

From Young's inequality, we further have

$$\|x^{k+1} - x^*\|^2 \leq \frac{8}{\mu^2} \left( \frac{1}{n} \sum_{i=1}^{n} \|\mathbf{H}_i^k - \nabla^2 f_i(x^*)\|_{\mathrm{F}} \|x^k - x^*\| \right)^2 + \frac{(L_* + 2L_{\mathrm{F}})^2}{2\mu^2} \|x^k - x^*\|^4$$

$$\leq \frac{8}{\mu^2} \|x^k - x^*\|^2 \left( \frac{1}{n} \sum_{i=1}^{n} \|\mathbf{H}_i^k - \nabla^2 f_i(x^*)\|_{\mathrm{F}}^2 \right) + \frac{(L_* + 2L_{\mathrm{F}})^2}{2\mu^2} \|x^k - x^*\|^4$$

$$= \frac{8}{\mu^2} \|x^k - x^*\|^2 \mathcal{H}^k + \frac{(L_* + 2L_{\mathrm{F}})^2}{2\mu^2} \|x^k - x^*\|^4, \tag{23}$$

where we use the convexity of $\|\cdot\|_{\mathrm{F}}^2$ in the second inequality.

Thus, from (22) and (23), we have the following unified bound for both *Option 1* and *Option 2*:

$$\|x^{k+1} - x^*\|^2 \leq \frac{C}{\mu^2} \|x^k - x^*\|^2 \mathcal{H}^k + \frac{D}{2\mu^2} \|x^k - x^*\|^4. \tag{24}$$

Assume $\|x^0 - x^*\|^2 \leq \frac{\mu^2}{2D}$ and $\mathcal{H}^k \leq \frac{\mu^2}{4C}$ for all $k \geq 0$. Then we show that $\|x^k - x^*\|^2 \leq \frac{\mu^2}{2D}$ for all $k \geq 0$ by induction. Assume $\|x^k - x^*\|^2 \leq \frac{\mu^2}{2D}$ for all $k \leq K$. Then from (24), we have

$$\|x^{K+1} - x^*\|^2 \leq \frac{1}{4} \|x^K - x^*\|^2 + \frac{1}{4} \|x^K - x^*\|^2 \leq \frac{\mu^2}{2D}.$$

Thus we have $\|x^k - x^*\|^2 \leq \frac{\mu^2}{2D}$ and $\mathcal{H}^k \leq \frac{\mu^2}{4C}$ for $k \geq 0$. Using (24) again, we obtain

$$\|x^{k+1} - x^*\|^2 \leq \frac{1}{2} \|x^k - x^*\|^2. \tag{25}$$

Assume $\|x^0 - x^*\|^2 \leq \frac{\mu^2}{2D}$ and $\mathcal{H}^k \leq \frac{\mu^2}{4C}$ for all $k \geq 0$. Then we show that $\|x^k - x^*\|^2 \leq \frac{\mu^2}{2D}$ for all $k \geq 0$ by induction. Assume $\|x^k - x^*\|^2 \leq \frac{\mu^2}{2D}$ for all $k \leq K$. Then from (24), we have

$$\|x^{K+1} - x^*\|^2 \leq \frac{1}{4} \|x^K - x^*\|^2 + \frac{1}{4} \|x^K - x^*\|^2 \leq \frac{\mu^2}{2D}.$$

Thus we have $\|x^k - x^*\|^2 \leq \frac{\mu^2}{2D}$ and $\mathcal{H}^k \leq \frac{\mu^2}{4C}$ for $k \geq 0$. Using (24) again, we obtain

$$\|x^{k+1} - x^*\|^2 \leq \frac{1}{2} \|x^k - x^*\|^2. \tag{26}$$

Thus, we derived the first rate of the theorem. Next, we invoke Lemma F.1 to have an upper bound for $\mathcal{H}^{k+1}$:

$$\mathbb{E}_k[\mathcal{H}^{k+1}] \leq \left( 1 - \frac{A}{2} \right) \mathcal{H}^k + \left( \frac{1}{A} + 3B \right) L_{\mathrm{F}}^2 \|x^k - x^*\|^2.$$

Using the above inequality and (26), for Lyapunov function $\Phi^k$ we deduce

$$\mathbb{E}_k[\Phi^{k+1}] \leq \left( 1 - \frac{A}{2} \right) \mathcal{H}^k + \left( \frac{1}{A} + 3B \right) L_{\mathrm{F}}^2 \|x^k - x^*\|^2 + 3 \left( \frac{1}{A} + 3B \right) L_{\mathrm{F}}^2 \|x^k - x^*\|^2$$

$$= \left( 1 - \frac{A}{2} \right) \mathcal{H}^k + \left( 1 - \frac{1}{3} \right) 6 \left( \frac{1}{A} + 3B \right) L_{\mathrm{F}}^2 \|x^k - x^*\|^2$$

$$\leq \left( 1 - \min \left\{ \frac{A}{2}, \frac{1}{3} \right\} \right) \Phi^k.$$

Hence $\mathbb{E}_k[\Phi^k] \leq \left(1 - \min\left\{\frac{A}{2}, \frac{1}{3}\right\}\right)^k \Phi^0$. Clearly, we further have $\mathbb{E}[\mathcal{H}^k] \leq \left(1 - \min\left\{\frac{A}{2}, \frac{1}{3}\right\}\right)^k \Phi^0$ and $\mathbb{E}[\|x^k - x^*\|^2] \leq \frac{A}{6(1+3AB)L_{\mathrm{F}}^2} \left(1 - \min\left\{\frac{A}{2}, \frac{1}{3}\right\}\right)^k \Phi^0$ for $k \geq 0$. Assume $x^k \neq x^*$ for all $k$. Then from (24), we have

$$\frac{\|x^{k+1} - x^*\|^2}{\|x^k - x^*\|^2} \leq \frac{C}{\mu^2}\mathcal{H}^k + \frac{D}{2\mu^2}\|x^k - x^*\|^2,$$

and by taking expectation, we have

$$\mathbb{E}\left[\frac{\|x^{k+1} - x^*\|^2}{\|x^k - x^*\|^2}\right] \leq \frac{C}{\mu^2}\mathbb{E}[\mathcal{H}^k] + \frac{D}{2\mu^2}\mathbb{E}[\|x^k - x^*\|^2]$$

$$\leq \left(1 - \min\left\{\frac{A}{2}, \frac{1}{3}\right\}\right)^k \left(C + \frac{AD}{12(1 + 3AB)L_{\mathrm{F}}^2}\right)\frac{\Phi^0}{\mu^2},$$

which concludes the proof.

### F.3  Proof of Lemma 4.3

We prove this by induction. Assume $\|\mathbf{H}_i^k - \nabla^2 f_i(x^*)\|_{\mathrm{F}}^2 \leq \frac{\mu^2}{4C}$ and $\|x^k - x^*\|^2 \leq e_1^2$ for $k \leq K$. Then we also have $\mathcal{H}^k \leq \frac{\mu^2}{4C}$ for $k \leq K$. From (24), we can get

$$\|x^{K+1} - x^*\|^2 \leq \frac{C}{\mu^2}\|x^K - x^*\|^2\mathcal{H}^K + \frac{D}{2\mu^2}\|x^K - x^*\|^4$$

$$\leq \frac{1}{4}\|x^K - x^*\|^2 + \frac{1}{4}\|x^K - x^*\|^2$$

$$\leq \|x^K - x^*\|^2 \leq e_1^2.$$

Using Lemma F.1 and the assumptions that we use non-random 3PC compressor, we have

$$\|\mathbf{H}_i^{K+1} - \nabla^2 f_i(x^*)\|_{\mathrm{F}}^2 \leq \left(1 - \frac{A}{2}\right)\|\mathbf{H}_i^K - \nabla^2 f_i(x^*)\|_{\mathrm{F}}^2 + \frac{1 + 3AB}{A}L_{\mathrm{F}}^2\|x^K - x^*\|^2$$

$$\leq \left(1 - \frac{A}{2}\right)\frac{\mu^2}{4C} + \frac{1 + 3AB}{A}L_{\mathrm{F}}^2 \cdot \frac{A^2\mu^2}{8(1 + 3AB)CL_{\mathrm{F}}^2}$$

$$= \frac{\mu^2}{4C}.$$

### F.4  Proof of Lemma 4.4

We prove this by induction. Assume $\|x^k - x^*\| \leq e_1$ and $\|\mathbf{H}_i^k - \nabla^2 f_i(x^*)\|_{\mathrm{F}}^2 \leq \frac{\mu^2}{4C}$ for $k \leq K$. Then we also have $\mathcal{H}^k \leq \frac{\mu^2}{4C}$ for $k \leq K$. From (24), we can get

$$\|x^{K+1} - x^*\|^2 \leq \frac{C}{\mu^2}\|x^K - x^*\|^2\mathcal{H}^K + \frac{D}{2\mu^2}\|x^K - x^*\|^4$$

$$\leq \frac{1}{4}\|x^K - x^*\|^2 + \frac{1}{4}\|x^K - x^*\|^2 \leq e_1^2.$$

From the definition

$$\mathbf{H}_i^{k+1} = \begin{cases} \mathbf{H}_i^k + \mathcal{C}(\nabla^2 f_i(x^{k+1}) - \mathbf{H}_i^k) & \text{with probability } p, \\ \mathbf{H}_i^k & \text{with probability } 1 - p. \end{cases} \tag{27}$$

we have two cases for $\mathbf{H}_i^{k+1}$ we need to upper bound individually instead of in expectation. Note that the case $\mathbf{H}_i^{k+1} = \mathbf{H}_i^k$ is trivial as $\|\mathbf{H}_i^{k+1} - \nabla^2 f_i(x^*)\|_{\mathrm{F}} = \|\mathbf{H}_i^k - \nabla^2 f_i(x^*)\|_{\mathrm{F}} \leq \frac{\mu}{2\sqrt{C}}$. For the other case when

$\mathbf{H}_i^{k+1} = \mathbf{H}_i^k + \mathcal{C}(\nabla^2 f_i(x^{k+1}) - \mathbf{H}_i^k)$, we have

$$
\begin{aligned}
& \|\mathbf{H}_i^{k+1} - \nabla^2 f_i(x^*)\|_{\mathrm{F}} \\
= {} & \|\mathbf{H}_i^k + \mathcal{C}(\nabla^2 f_i(x^{k+1}) - \mathbf{H}_i^k) - \nabla^2 f_i(x^*)\|_{\mathrm{F}} \\
\leq {} & \|\mathcal{C}(\nabla^2 f_i(x^{k+1}) - \mathbf{H}_i^k) - (\nabla^2 f_i(x^{k+1}) - \mathbf{H}_i^k)\|_{\mathrm{F}} + \|\nabla^2 f_i(x^{k+1}) - \nabla^2 f_i(x^*)\|_{\mathrm{F}} \\
\leq {} & \sqrt{1-\alpha}\|\nabla^2 f_i(x^{k+1}) - \mathbf{H}_i^k\|_{\mathrm{F}} + L_{\mathrm{F}}\|x^{k+1} - x^*\| \\
\leq {} & \sqrt{1-\alpha}\|\mathbf{H}_i^k - \nabla^2 f_i(x^*)\|_{\mathrm{F}} + \sqrt{1-\alpha}\|\nabla^2 f_i(x^{k+1}) - \nabla^2 f_i(x^*)\|_{\mathrm{F}} + L_{\mathrm{F}}\|x^{k+1} - x^*\| \\
\leq {} & \sqrt{1-\alpha}\|\mathbf{H}_i^k - \nabla^2 f_i(x^*)\|_{\mathrm{F}} + 2L_{\mathrm{F}}\|x^{k+1} - x^*\| \\
\leq {} & \sqrt{1-\alpha}\frac{\mu}{2\sqrt{C}} + 2L_{\mathrm{F}} \cdot \frac{(1-\sqrt{1-\alpha})\mu}{4\sqrt{C}L_{\mathrm{F}}} = \frac{\mu}{2\sqrt{C}},
\end{aligned}
$$

which completes our induction step and the proof.

# G   Deferred Proofs from Section 5 (Newton-3PC-BC)

## G.1   Proof of Theorem 5.1

First we have

$$
\begin{aligned}
\|x^{k+1} - x^*\|^2 &= \|z^k - x^* - [\mathbf{H}^k]_\mu^{-1} g^k\|^2 \\
&= \left\| [\mathbf{H}^k]_\mu^{-1} \left( [\mathbf{H}^k]_\mu(z^k - x^*) - (g^k - \nabla f(x^*)) \right) \right\|^2 \\
&\leq \frac{1}{\mu^2} \left\| [\mathbf{H}^k]_\mu(z^k - x^*) - (g^k - \nabla f(x^*)) \right\|^2,
\end{aligned}
\tag{28}
$$

where we use $\nabla f(x^*) = 0$ in the second equality, and $\|[\mathbf{H}^k]_\mu^{-1}\| \leq \frac{1}{\mu}$ in the last inequality.

If $\xi^k = 1$, then

$$
\begin{aligned}
& \left\| [\mathbf{H}^k]_\mu(z^k - x^*) - (g^k - \nabla f(x^*)) \right\|^2 \\
= {} & \left\| \nabla f(z^k) - \nabla f(x^*) - \nabla^2 f(x^*)(z^k - x^*) + (\nabla^2 f(x^*) - [\mathbf{H}^k]_\mu)(z^k - x^*) \right\|^2 \\
\leq {} & 2\left\| \nabla f(z^k) - \nabla f(x^*) - \nabla^2 f(x^*)(z^k - x^*) \right\|^2 + 2\left\| (\nabla^2 f(x^*) - [\mathbf{H}^k]_\mu)(z^k - x^*) \right\|^2 \\
\leq {} & \frac{L_*^2}{2}\|z^k - x^*\|^4 + 2\|[\mathbf{H}^k]_\mu - \nabla^2 f(x^*)\|^2 \cdot \|z^k - x^*\|^2 \\
\leq {} & \frac{L_*^2}{2}\|z^k - x^*\|^4 + 2\|\mathbf{H}^k - \nabla^2 f(x^*)\|_{\mathrm{F}}^2\|z^k - x^*\|^2 \\
= {} & \frac{L_*^2}{2}\|z^k - x^*\|^4 + 2\left\| \frac{1}{n}\mathbf{H}_i^k - \frac{1}{n}\nabla^2 f_i(x^*) \right\|_{\mathrm{F}}^2 \|z^k - x^*\|^2 \\
\leq {} & \frac{L_*^2}{2}\|z^k - x^*\|^4 + \frac{2}{n}\sum_{i=1}^n \|\mathbf{H}_i^k - \nabla^2 f_i(x^*)\|_{\mathrm{F}}^2\|z^k - x^*\|^2,
\end{aligned}
\tag{29}
$$

where in the second inequality, we use the Lipschitz continuity of the Hessian of $f$, and in the last inequality, we use the convexity of $\|\cdot\|_{\mathrm{F}}^2$.

If $\xi^k = 0$, then

$$
\begin{aligned}
& \left\| [\mathbf{H}^k]_\mu (z^k - x^*) - (g^k - \nabla f(x^*)) \right\|^2 \\
&= \left\| [\mathbf{H}^k]_\mu (z^k - w^k) + \nabla f(w^k) - \nabla f(x^*) - [\mathbf{H}^k]_\mu (z^k - x^*) \right\|^2 \\
&= \left\| [\mathbf{H}^k]_\mu (x^* - w^k) + \nabla f(w^k) - \nabla f(x^*) \right\|^2 \\
&= \left\| \nabla f(w^k) - \nabla f(x^*) - \nabla^2 f(x^*)(w^k - x^*) + (\nabla^2 f(x^*) - [\mathbf{H}^k]_\mu)(w^k - x^*) \right\|^2 \\
&\leq \frac{L_*^2}{2} \|w^k - x^*\|^4 + 2\|\mathbf{H}^k - \nabla^2 f(x^*)\|_{\mathrm{F}}^2 \|w^k - x^*\|^2 \\
&\leq \frac{L_*^2}{2} \|w^k - x^*\|^4 + \frac{2}{n} \sum_{i=1}^n \|\mathbf{H}_i^k - \nabla^2 f_i(x^*)\|_{\mathrm{F}}^2 \|w^k - x^*\|^2.
\end{aligned}
\tag{30}
$$

For $k \geq 1$, from the above three inequalities, we can obtain

$$
\begin{aligned}
\mathbb{E}_k \|x^{k+1} - x^*\|^2 &\leq \frac{L_*^2 p}{2\mu^2} \|z^k - x^*\|^4 + \frac{2p}{n\mu^2} \sum_{i=1}^n \|\mathbf{H}_i^k - \nabla^2 f_i(x^*)\|_{\mathrm{F}}^2 \|z^k - x^*\|^2 \\
&\quad + \frac{L_*^2 (1-p)}{2\mu^2} \|w^k - x^*\|^4 + \frac{2(1-p)}{n\mu^2} \sum_{i=1}^n \|\mathbf{H}_i^k - \nabla^2 f_i(x^*)\|_{\mathrm{F}}^2 \|w^k - x^*\|^2 \\
&= \frac{p}{2\mu^2} \left( L_*^2 \|z^k - x^*\|^2 + 4\mathcal{H}^k \right) \|z^k - x^*\|^2 \\
&\quad + \frac{(1-p)}{2\mu^2} \left( L_*^2 \|w^k - x^*\|^2 + 4\mathcal{H}^k \right) \|w^k - x^*\|^2,
\end{aligned}
\tag{31}
$$

where we denote $\mathcal{H}^k := \frac{1}{n} \sum_{i=1}^n \|\mathbf{H}_i^k - \nabla^2 f_i(x^*)\|_{\mathrm{F}}^2$.

For $k = 0$, since $z^0 = w^0$, it is easy to verify that the above equality also holds.

From the update rule of $z^k$, we have

$$
\begin{aligned}
\mathbb{E}_k \|z^{k+1} - x^*\|^2 &\leq (1+\alpha) \mathbb{E}_k \|z^{k+1} - x^{k+1}\|^2 + \left(1 + \frac{1}{\alpha}\right) \mathbb{E}_k \|x^{k+1} - x^*\|^2 \\
&\leq (1+\alpha)(1 - A_M) \|z^k - x^k\|^2 + (1+\alpha) B_M \mathbb{E}_k \|x^{k+1} - x^k\|^2 + \left(1 + \frac{1}{\alpha}\right) \mathbb{E}_k \|x^{k+1} - x^*\|^2 \\
&\leq (1+\alpha)(1 - A_M)(1 + \beta) \|z^k - x^*\|^2 + (1+\alpha)(1 - A_M)\left(1 + \frac{1}{\beta}\right) \|x^k - x^*\|^2 \\
&\quad + 2(1+\alpha) B_M \|x^k - x^*\|^2 + \left(2(1+\alpha) B_M + 1 + \frac{1}{\alpha}\right) \mathbb{E}_k \|x^{k+1} - x^*\|^2,
\end{aligned}
$$

for any $\alpha > 0$, $\beta > 0$. By choosing $\alpha = \frac{A_M}{4}$ and $\beta = \frac{A_M}{4(1 - \frac{3A_M}{4})}$, we arrive at

$$
\begin{aligned}
\mathbb{E}_k \|z^{k+1} - x^*\|^2 &\leq \left(1 - \frac{A_M}{2}\right) \|z^k - x^*\|^2 + \left(\frac{4}{A_M} - 3 + \frac{5B_M}{2}\right) \|x^k - x^*\|^2 + \left(\frac{4}{A_M} + 1 + \frac{5B_M}{2}\right) \mathbb{E}_k \|x^{k+1} - x^*\|^2 \\
&\leq \left(1 - \frac{A_M}{2}\right) \|z^k - x^*\|^2 + C_M \|x^k - x^*\|^2 + C_M \mathbb{E}_k \|x^{k+1} - x^*\|^2,
\end{aligned}
\tag{32}
$$

where we denote $C_M := \frac{4}{A_M} + 1 + \frac{5B_M}{2}$. Then we have

$$\mathbb{E}_k[\|z^{k+1} - x^*\|^2 + 2C_M\|x^{k+1} - x^*\|^2] \leq \left(1 - \frac{A_M}{2}\right)\|z^k - x^*\|^2 + C_M\|x^k - x^*\|^2 + 3C_M\mathbb{E}_k\|x^{k+1} - x^*\|^2$$

$$\overset{(31)}{\leq} \left(1 - \frac{A_M}{2}\right)\|z^k - x^*\|^2 + \frac{3C_M p}{2\mu^2}\left(L_*^2\|z^k - x^*\|^2 + 4\mathcal{H}^k\right)\|z^k - x^*\|^2$$

$$+ \frac{3C_M(1-p)}{2\mu^2}\left(L_*^2\|w^k - x^*\|^2 + 4\mathcal{H}^k\right)\|w^k - x^*\|^2 + C_M\|x^k - x^*\|^2.$$

Assume $\|z^k - x^*\|^2 \leq \frac{A_M \mu^2}{24 C_M L_*^2}$ and $\mathcal{H}^k \leq \frac{A_M \mu^2}{96 C_M}$ for $k \geq 0$. Then from the update rule of $w^k$, we also have $\|w^k - x^*\|^2 \leq \frac{A_M \mu^2}{24 C_M L_*^2}$ for $k \geq 0$. Therefore, we have

$$\mathbb{E}_k[\|z^{k+1} - x^*\|^2 + 2C_M\|x^{k+1} - x^*\|^2] \leq \left(1 - \frac{A_M}{2} + \frac{A_M p}{8}\right)\|z^k - x^*\|^2 + \frac{A_M(1-p)}{8}\|w^k - x^*\|^2 + C_M\|x^k - x^*\|^2. \tag{33}$$

From the update rule of $w^k$, we have

$$\mathbb{E}_k\|w^{k+1} - x^*\|^2 = p\|z^{k+1} - x^*\|^2 + (1-p)\|w^k - x^*\|^2. \tag{34}$$

Define $\Phi_1^k := \|z^k - x^*\|^2 + C_M\|x^k - x^*\|^2 + \frac{A_M(1-p)}{4p}\|w^k - x^*\|^2$. Then we have

$$\mathbb{E}_k[\Phi_1^{k+1}] = \mathbb{E}_k[\|z^{k+1} - x^*\|^2 + 2C_M\|x^{k+1} - x^*\|^2] + \frac{A_M(1-p)}{4p}\mathbb{E}_k\|w^{k+1} - x^*\|^2$$

$$\overset{(34)}{\leq} \left(1 + \frac{A_M(1-p)}{4}\right)\mathbb{E}_k[\|z^{k+1} - x^*\|^2 + 2C_M\|x^{k+1} - x^*\|^2] + \frac{A_M(1-p)^2}{4p}\|w^k - x^*\|^2$$

$$\overset{(33)}{\leq} \left(1 + \frac{A_M(1-p)}{4}\right)\left(1 - \frac{A_M}{2} + \frac{A_M p}{8}\right)\|z^k - x^*\|^2 + \left(1 + \frac{A_M(1-p)}{4}\right)C_M\|x^k - x^*\|^2$$

$$+ \left(\left(1 + \frac{A_M(1-p)}{4}\right)\frac{A_M(1-p)}{8} + \frac{A_M(1-p)^2}{4p}\right)\|w^k - x^*\|^2$$

$$\leq \left(1 - \frac{A_M}{4}\right)\|z^k - x^*\|^2 + \left(1 - \frac{3}{8}\right)2C_M\|x^k - x^*\|^2 + \frac{A_M(1-p)}{4p}\left(1 - \frac{3p}{8}\right)\|w^k - x^*\|^2$$

$$\leq \left(1 - \frac{\min\{2A_M, 3p\}}{8}\right)\Phi_1^k.$$

By applying the tower property, we have

$$\mathbb{E}[\Phi_1^{k+1}] \leq \left(1 - \frac{\min\{2A_M, 3p\}}{8}\right)\mathbb{E}[\Phi_1^k].$$

Unrolling the recursion, we can get the result.

### G.2   Proof of Lemma 5.2

We prove the results by mathematical induction. Assume the results hold for $k \leq K$. From the update rule of $w^k$, we know $\|w^k - x^*\|^2 \leq \min\{\frac{A_M \mu^2}{24 C_M L_*^2}, \frac{A_W A_M \mu^2}{384 C_M C_W L_F^2}\}$ for $k \leq K$. If $\xi^K = 1$, from (28) and (29), we have

$$\|x^{K+1} - x^*\|^2 \leq \frac{1}{\mu^2}\left(\frac{L_*^2}{2}\|z^K - x^*\|^2 + 2\mathcal{H}^K\right)\|z^K - x^*\|^2 \tag{35}$$

$$\leq \frac{A_M}{24 C_M}\|z^K - x^*\|^2.$$

If $\xi^K = 0$, from $\|w^K - x^*\|^2 \leq \min\{\frac{A_M\mu^2}{24C_ML_*^2}, \frac{A_WA_M\mu^2}{384C_MC_WL_F^2}\}$ and (30), we can obtain the above inequality similarly. From the upper bound of $\|z^K - x^*\|^2$, we further have $\|x^{K+1} - x^*\| \leq \frac{11A_M}{24C_M}\min\{\frac{A_M\mu^2}{24C_M^2L_*^2}, \frac{A_WA_M\mu^2}{384C_MC_WL_F^2}\}$. Then from (32) and the fact that $\mathcal{C}_{z^k,x^k}^M(x^{k+1})$ is deterministic, we have

$$
\begin{aligned}
\|z^{K+1} - x^*\|^2 &\leq \left(1 - \frac{A_M}{2}\right)\|z^K - x^*\|^2 + C_M\|x^K - x^*\|^2 + C_M\|x^{K+1} - x^*\|^2 \\
&\leq \left(1 - \frac{A_M}{2} + \frac{A_M}{24}\right)\|z^K - x^*\|^2 + C_M \cdot \frac{11A_M}{24C_M}\min\{\frac{A_M\mu^2}{24C_M^2L_*^2}, \frac{A_WA_M\mu^2}{384C_MC_WL_F^2}\} \\
&\leq \min\{\frac{A_M\mu^2}{24C_M^2L_*^2}, \frac{A_WA_M\mu^2}{384C_MC_WL_F^2}\}.
\end{aligned}
$$

For $\|\mathbf{H}_i^{k+1} - \nabla^2 f_i(x^*)\|_F^2$, we have

$$
\begin{aligned}
\mathbb{E}_k\|\mathbf{H}_i^{k+1} - \nabla^2 f_i(x^*)\|_F^2 &\leq (1+\alpha)\mathbb{E}_k\|\mathbf{H}_i^k - \nabla^2 f_i(z^{k+1})\|_F^2 + \left(1 + \frac{1}{\alpha}\right)\mathbb{E}_k\|\nabla^2 f_i(z^{k+1}) - \nabla^2 f_i(x^*)\|_F^2 \\
&\leq (1+\alpha)(1 - A_W)\|\mathbf{H}_i^k - \nabla^2 f_i(z^k)\|_F^2 + (1+\alpha)B_W\mathbb{E}_k\|\nabla^2 f_i(z^k) - \nabla^2 f_i(z^{k+1})\|_F^2 \\
&\quad + \left(1 + \frac{1}{\alpha}\right)\mathbb{E}_k\|\nabla^2 f_i(z^{k+1}) - \nabla^2 f_i(x^*)\|_F^2 \\
&\leq (1+\alpha)(1 - A_W)\|\mathbf{H}_i^k - \nabla^2 f_i(z^k)\|_F^2 + (1+\alpha)B_WL_F^2\mathbb{E}_k\|z^k - z^{k+1}\|^2 \\
&\quad + \left(1 + \frac{1}{\alpha}\right)L_F^2\mathbb{E}_k\|z^{k+1} - x^*\|^2 \\
&\leq (1+\alpha)(1 - A_W)(1+\beta)\|\mathbf{H}_i^k - \nabla^2 f_i(x^*)\|_F^2 + (1+\alpha)(1 - A_W)\left(1 + \frac{1}{\beta}\right)L_F^2\|z^k - x^*\|^2 \\
&\quad + 2(1+\alpha)B_WL_F^2\|z^k - x^*\|^2 + \left(2(1+\alpha)B_W + 1 + \frac{1}{\alpha}\right)L_F^2\|z^{k+1} - x^*\|^2,
\end{aligned}
$$

for any $\alpha > 0$, $\beta > 0$. By choosing $\alpha = \frac{A_W}{4}$ and $\beta = \frac{A_W}{4(1 - \frac{3A_W}{4})}$, we arrive at

$$
\mathbb{E}_k\|\mathbf{H}_i^{k+1} - \nabla^2 f_i(x^*)\|_F^2 \leq \left(1 - \frac{A_W}{2}\right)\|\mathbf{H}_i^k - \nabla^2 f_i(x^*)\|_F^2 + C_WL_F^2\|z^k - x^*\|^2 + C_WL_F^2\mathbb{E}_k\|z^{k+1} - x^*\|^2, \quad (36)
$$

where we denote $C_W := \frac{4}{A_W} + 1 + \frac{5B_W}{2}$. Since $\mathcal{C}_{\mathbf{H}_i^k, \nabla^2 f_i(z^k)}^W(z^{k+1})$ is disterministic, from (36), we have

$$
\begin{aligned}
\mathcal{H}^{K+1} &\leq \left(1 - \frac{A_W}{2}\right)\mathcal{H}^K + C_WL_F^2\|z^K - x^*\|^2 + C_WL_F^2\|z^{K+1} - x^*\|^2 \\
&\leq \left(1 - \frac{A_W}{2}\right)\frac{A_M\mu^2}{96C_M} + 2C_WL_F^2 \cdot \frac{A_WA_M\mu^2}{384C_MC_WL_F^2} \\
&\leq \frac{A_M\mu^2}{96C_M}.
\end{aligned}
$$

## G.3 Proof of Lemma 5.3

We prove the results by mathematical induction. From the assumption on $\mathbf{H}_i^k$, we have

$$
\begin{aligned}
\mathcal{H}^k &= \frac{1}{n}\sum_{i=1}^n \|\mathbf{H}_i^k - \nabla^2 f_i(x^*)\|^2 \\
&\leq \frac{1}{n}\sum_{i=1}^n d^2\max_{jl}\{|(\mathbf{H}_i^k)_{jl} - (\nabla^2 f(x^*))_{jl}|^2\} \\
&\leq d^2L_\infty^2\max_{0\leq t\leq k}\|z^t - x^*\|^2. \quad (37)
\end{aligned}
$$

Then from $\|x^0 - x^*\|^2 \leq \tilde{c}_1$, we have $\mathcal{H}^0 \leq \min\{\frac{A_M \mu^2}{96 C_M}, \frac{\mu^2}{4d}\}$. Assume the results hold for all $k \leq K$. If $\xi^K = 1$, from (35), we have

$$\|x^{K+1} - x^*\|^2 \leq \frac{1}{\mu^2} \left( \frac{L_*^2}{2} \|z^K - x^*\|^2 + 2\mathcal{H}^K \right) \|z^K - x^*\|^2$$
$$\leq \frac{1}{d} \|z^K - x^*\|^2$$
$$\leq \tilde{c}_1.$$

If $\xi^K = 0$, from $\|w^K - x^*\|^2 \leq d\tilde{c}_1$ and (30), we can obtain the above inequality similarly. From the assumption on $z^k$, we have

$$\|z^{K+1} - x^*\|^2 \leq d \max_j |z_j^{K+1} - x_j^*|^2$$
$$\leq d \max_{0 \leq t \leq K+1} \|x^t - x^*\|^2$$
$$\leq d\tilde{c}_1.$$

At last, using (37), we can get $\mathcal{H}^{K+1} \leq \min\{\frac{A_M \mu^2}{96 C_M}, \frac{\mu^2}{4d}\}$, which completes the proof.

## H  Extension to Bidirectional Compression and Partial Participation

In this section, we unify the bidirectional compression and partial participation in Algorithm 3. The algorithm can also be regarded as an extension of BL2 in (Qian et al., 2022) by the three point compressor. Here the symmetrization operator $[\cdot]_s$ is defined as

$$[\mathbf{A}]_s := \frac{\mathbf{A} + \mathbf{A}^\top}{2}$$

for any $\mathbf{A} \in \mathbb{R}^{d \times d}$. The update of the global model at $k$-th iteration is

$$x^{k+1} = \left([\mathbf{H}^k]_s + l^k \mathbf{I}\right)^{-1} g^k,$$

where $\mathbf{H}^k$, $l^k$, and $g^k$ are the average of $\mathbf{H}_i^k$, $l_i^k$, and $g_i^k$ respectively. This update is based on the following step in Stochastic Newton method (Kovalev et al., 2019)

$$x^{k+1} = \left[ \frac{1}{n} \sum_{i=1}^n \nabla^2 f_i(w_i^k) \right]^{-1} \left[ \frac{1}{n} \sum_{i=1}^n \left( \nabla^2 f_i(w_i^k) w_i^k - \nabla f_i(w_i^k) \right) \right].$$

We use $[\mathbf{H}_i^k]_s + l_i^k \mathbf{I}$ to estimate $\nabla^2 f_i(w_i^k)$, and $g_i^k$ to estimate $\nabla^2 f_i(w_i^k) w_i^k - \nabla f_i(w_i^k)$, where $l_i^k = \|[\mathbf{H}_i^k]_s - \nabla^2 f_i(z_i^k)\|_F$ is adopted to guarantee the positive definiteness of $[\mathbf{H}^k]_s + l^k \mathbf{I}$. Hence, like BL2 in (Qian et al., 2022), we maintain the key relation

$$g_i^k = ([\mathbf{H}_i^k]_s + l_i^k \mathbf{I}) w_i^k - \nabla f_i(w_i^k). \tag{38}$$

Since each node has a local model $w_i^k$, we introduce $z_i^k$ to apply the bidirectional compression with the three point compressor and $\mathbf{H}_i^k$ is expected to learn $h^i(\nabla^2 f_i(z_i^k))$ iteratively. For the update of $g_i^k$ on the server when $\xi_i^k = 0$, from (38), it is natural to let

$$g_i^{k+1} - g_i^k = ([\mathbf{H}_i^{k+1}]_s - [\mathbf{H}_i^k]_s + l_i^{k+1} \mathbf{I} - l_i^k \mathbf{I}) w_i^{k+1},$$

since we have $w_i^{k+1} = w_i^k$ when $\xi_i^k = 0$. The convergence results of Newton-3PC-BC-PP are stated in the following two theorems.

---

**Algorithm 3** Newton-3PC-BC-PP (Newton's method with 3PC, BC and  Partial Participation)

---

1: **Parameters:** Worker's ($\mathcal{C}^W$) and Master's ($\mathcal{C}^M$) 3PC; probability $p \in (0, 1]$; $0 < \tau \leq n$

2: **Initialization:** $w_i^0 = z_i^0 = x^0 \in \mathbb{R}^d$; $\mathbf{H}_i^0 \in \mathbb{R}^{d \times d}$; $l_i^0 = \|[\mathbf{H}_i^0]_s - \nabla^2 f_i(w_i^0)\|_F$; $g_i^0 = ([\mathbf{H}_i^0]_s + l_i^0 \mathbf{I}) w_i^0 - \nabla f_i(w_i^0)$; Moreover: $\mathbf{H}^0 = \frac{1}{n} \sum_{i=1}^n \mathbf{H}_i^0$; $l^0 = \frac{1}{n} \sum_{i=1}^n l_i^0$; $g^0 = \frac{1}{n} \sum_{i=1}^n g_i^0$

3: **on** server

4:      $x^{k+1} = ([\mathbf{H}^k]_s + l^k \mathbf{I})^{-1} g^k$,

5:      choose a subset $S^k \subseteq [n]$ such that $\mathbb{P}[i \in S^k] = \tau/n$ for all $i \in [n]$

6:      $z_i^{k+1} = \mathcal{C}_{z_i^k, x^k}^M(x^{k+1})$ for $i \in S^k$

7:      $z_i^{k+1} = z_i^k$,     $w_i^{k+1} = w_i^k$ for $i \notin S^k$

8:      Send $\mathcal{C}_{z_i^k, x^k}^M(x^{k+1})$ to  the selected devices $i \in S^k$

9: **for** each device $i = 1, \ldots, n$ in parallel **do**

10:      **for participating devices** $i \in S^k$ **do**

11:        $z_i^{k+1} = \mathcal{C}_{z_i^k, x^k}^M(x^{k+1})$

12:        $\mathbf{H}_i^{k+1} = \mathcal{C}_{\mathbf{H}_i^k, \nabla^2 f_i(z_i^k)}^W(\nabla^2 f_i(z_i^{k+1}))$

13:        $l_i^{k+1} = \|[\mathbf{H}_i^{k+1}]_s - \nabla^2 f_i(z_i^{k+1})\|_F$

14:        Sample $\xi_i^{k+1} \sim \text{Bernoulli}(p)$

15:        **if** $\xi_i^{k+1} = 1$

16:          $w_i^{k+1} = z_i^{k+1}$, $g_i^{k+1} = ([\mathbf{H}_i^{k+1}]_s + l_i^{k+1} \mathbf{I}) w_i^{k+1} - \nabla f_i(w_i^{k+1})$, send $g_i^{k+1} - g_i^k$ to server

17:        **if** $\xi_i^{k+1} = 0$

18:          $w_i^{k+1} = w_i^k$, $g_i^{k+1} = ([\mathbf{H}_i^{k+1}]_s + l_i^{k+1} \mathbf{I}) w_i^{k+1} - \nabla f_i(w_i^{k+1})$

19:        Send $\mathbf{H}_i^{k+1}$, $l_i^{k+1} - l_i^k$, and $\xi_i^{k+1}$ to the server

20:      **for non-participating devices** $i \notin S^k$ **do**

21:        $z_i^{k+1} = z_i^k$, $w_i^{k+1} = w_i^k$, $\mathbf{H}_i^{k+1} = \mathbf{H}_i^k$, $l_i^{k+1} = l_i^k$, $g_i^{k+1} = g_i^k$

22: **end for**

23: **on** server

24:      **if** $\xi_i^{k+1} = 1$

25:        $w_i^{k+1} = z_i^{k+1}$, receive $g_i^{k+1} - g_i^k$

26:      **if** $\xi_i^{k+1} = 0$

27:        $w_i^{k+1} = w_i^k$, $g_i^{k+1} - g_i^k = [\mathbf{H}_i^{k+1} - \mathbf{H}_i^k]_s w_i^{k+1} + (l_i^{k+1} - l_i^k) w_i^{k+1}$

28:      $g^{k+1} = g^k + \frac{1}{n} \sum_{i \in S^k} (g_i^{k+1} - g_i^k)$

29:      $\mathbf{H}^{k+1} = \frac{1}{n} \sum_{i=1}^n \mathbf{H}_i^{k+1}$

30:      $l^{k+1} = l^k + \frac{1}{n} \sum_{i \in S^k} (l_i^{k+1} - l_i^k)$

---

For $k \geq 0$, define Lyapunov function

$$\Phi_3^k := \mathcal{Z}^k + \frac{2\tau C_M}{n} \|x^k - x^*\|^2 + \frac{A_M}{4p} \mathcal{W}^k,$$

where $\tau \in [n]$ is the number of devices participating in each round.

**Theorem H.1.** *Let Assumption 4.1. Assume* $\|z_i^k - x^*\|^2 \leq \frac{A_M \mu^2}{36(H^2 + 4L_F^2) C_M}$ *and* $\mathcal{H}^k \leq \frac{A_M \mu^2}{576 C_M}$ *for all* $i \in [n]$ *and* $k \geq 0$. *Then we have*

$$\mathbb{E}[\Phi_3^k] \leq \left(1 - \frac{\tau \min\{2A_M, 3p\}}{8n}\right)^k \Phi_3^0,$$

*for* $k \geq 0$.

*Proof.* First, similar to (30) in (Qian et al., 2022), we can get

$$\|x^{k+1} - x^*\|^2 \leq \frac{3L_*^2}{4\mu^2}(\mathcal{W}^k)^2 + \frac{12\mathcal{W}^k}{n\mu^2}\sum_{i=1}^n \|\mathbf{H}_i^k - \nabla^2 f_i(x^*)\|_F^2 + \frac{3L_F^2}{\mu^2}\mathcal{Z}^k\mathcal{W}^k$$

$$= \frac{3L_*^2}{4\mu^2}(\mathcal{W}^k)^2 + \frac{12\mathcal{W}^k}{\mu^2}\mathcal{H}^k + \frac{3L_F^2}{\mu^2}\mathcal{Z}^k\mathcal{W}^k, \tag{39}$$

where $\mathcal{W}^k = \frac{1}{n}\sum_{i=1}^n \|w_i^k - x^*\|^2$ and $\mathcal{Z}^k = \frac{1}{n}\sum_{i=1}^n \|z_i^k - x^*\|^2$. For $i \in S^k$, we have $z_i^{k+1} = \mathcal{C}_{z_i^k, x^k}^M(x^{k+1})$.
Then, similar to (32), we have

$$\mathbb{E}_k\|z_i^{k+1} - x^*\|^2 \leq \left(1 - \frac{A_M}{2}\right)\|z_i^k - x^*\|^2 + C_M\|x^k - x^*\|^2 + C_M\|x^{k+1} - x^*\|^2.$$

Noticing that $\mathbb{P}[i \in S^k] = \tau/n$ and $z_i^{k+1} = z_i^k$ for $i \notin S^k$, we further have

$$\mathbb{E}_k\|z_i^{k+1} - x^*\|^2 = \frac{\tau}{n}\mathbb{E}_k[\|z_i^{k+1} - x^*\|^2 \mid i \in S^k] + \left(1 - \frac{\tau}{n}\right)\mathbb{E}_k[\|z_i^{k+1} - x^*\|^2 \mid i \notin S^k]$$

$$\leq \frac{\tau}{n}\left(1 - \frac{A_M}{2}\right)\|z_i^k - x^*\|^2 + \frac{\tau C_M}{n}\|x^k - x^*\|^2 + \frac{\tau C_M}{n}\|x^{k+1} - x^*\|^2 + \left(1 - \frac{\tau}{n}\right)\|z_i^k - x^*\|^2$$

$$= \left(1 - \frac{\tau A_M}{2n}\right)\|z_i^k - x^*\|^2 + \frac{\tau C_M}{n}\|x^k - x^*\|^2 + \frac{\tau C_M}{n}\|x^{k+1} - x^*\|^2,$$

which implies that

$$\mathbb{E}_k[\mathcal{Z}^{k+1}] = \frac{1}{n}\sum_{i=1}^n \mathbb{E}_k\|z_i^{k+1} - x^*\|^2$$

$$\leq \frac{1}{n}\sum_{i=1}^n \left(1 - \frac{\tau A_M}{2n}\right)\|z_i^k - x^*\|^2 + \frac{\tau C_M}{n}\|x^k - x^*\|^2 + \frac{\tau C_M}{n}\|x^{k+1} - x^*\|^2$$

$$= \left(1 - \frac{\tau A_M}{2n}\right)\mathcal{Z}^k + \frac{\tau C_M}{n}\|x^k - x^*\|^2 + \frac{\tau C_M}{n}\|x^{k+1} - x^*\|^2. \tag{40}$$

Combining (39) and (40), we have

$$\mathbb{E}_k[\mathcal{Z}^{k+1} + \frac{2\tau C_M}{n}\|x^{k+1} - x^*\|^2]$$

$$\leq \left(1 - \frac{\tau A_M}{2n}\right)\mathcal{Z}^k + \frac{\tau C_M}{n}\|x^k - x^*\|^2 + \frac{3\tau C_M}{n}\|x^{k+1} - x^*\|^2$$

$$\leq \left(1 - \frac{\tau A_M}{2n}\right)\mathcal{Z}^k + \frac{\tau C_M}{n}\|x^k - x^*\|^2 + \frac{3\tau C_M}{n}\left(\frac{3L_*^2}{4\mu^2}\mathcal{W}^k + \frac{12\mathcal{H}^k}{\mu^2} + \frac{3L_F^2\mathcal{Z}^k}{\mu^2}\right)\mathcal{W}^k.$$

Assume $\|z_i^k - x^*\|^2 \leq \frac{A_M\mu^2}{36(L_*^2 + 4L_F^2)C_M}$ and $\mathcal{H}^k \leq \frac{A_M\mu^2}{576C_M}$ for all $i \in [n]$ and $k \geq 0$. Then we have

$$\frac{3L_*^2}{4\mu^2}\mathcal{W}^k + \frac{12\mathcal{H}^k}{\mu^2} + \frac{3L_F^2\mathcal{Z}^k}{\mu^2} \leq \frac{A_M}{24C_M},$$

which indicates that

$$\mathbb{E}_k[\mathcal{Z}^{k+1} + \frac{2\tau C_M}{n}\|x^{k+1} - x^*\|^2] \leq \left(1 - \frac{\tau A_M}{2n}\right)\mathcal{Z}^k + \frac{\tau C_M}{n}\|x^k - x^*\|^2 + \frac{\tau A_M}{8n}\mathcal{W}^k. \tag{41}$$

For $\mathcal{W}^k$, similar to (32) in (Qian et al., 2022), we have

$$\mathbb{E}_k[\mathcal{W}^{k+1}] = \left(1 - \frac{\tau p}{n}\right)\mathcal{W}^k + \frac{\tau p}{n}\mathbb{E}[\mathcal{Z}^{k+1}].$$

Then from the above two inequalities we have

$$
\begin{aligned}
&\mathbb{E}_k[\Phi_3^{k+1}] \\
&\leq \left(1 + \frac{\tau A_M}{4n}\right) \mathbb{E}_k[\mathcal{Z}^{k+1} + \frac{2\tau C_M}{n}\|x^{k+1} - x^*\|^2] + \frac{A_M}{4p}\left(1 - \frac{\tau p}{n}\right)\mathcal{W}^k \\
&\overset{(41)}{\leq} \left(1 - \frac{\tau A_M}{4n}\right)\mathcal{Z}^k + \left(1 + \frac{\tau A_M}{4n}\right)\frac{\tau C_M}{n}\|x^k - x^*\|^2 + \frac{A_M}{4p}\left(1 - \frac{\tau p}{n} + \frac{\tau p}{2n}\left(1 + \frac{\tau A_M}{4n}\right)\right)\mathcal{W}^k \\
&\leq \left(1 - \frac{\tau\min\{2A_M, 3p\}}{8n}\right)\Phi_3^k.
\end{aligned}
$$

By applying the tower property, we have

$$
\mathbb{E}[\Phi_3^{k+1}] \leq \left(1 - \frac{\tau\min\{2A_M, 3p\}}{8n}\right)\mathbb{E}[\Phi_3^k].
$$

Unrolling the recursion, we can obtain the result.

$\square$

Define $\Phi_4^k = \mathcal{H}^k + \frac{16C_W L_{\mathrm{F}}^2}{A_M}\|x^k - x^*\|^2$ for $k \geq 0$, where $C_W := \frac{4}{A} + 1 + \frac{5B}{2}$.

**Theorem H.2.** *Let Assumption 4.1 holds, $\xi^k \equiv 1$, $S^k \equiv [n]$, and $\mathcal{C}_{z_i^k, x^k}^M(x^{k+1}) \equiv x^{k+1}$ for all $i \in [n]$ and $k \geq 0$. Assume $\|z_i^k - x^*\|^2 \leq \frac{A_M\mu^2}{36(L_*^2 + 4L_{\mathrm{F}}^2)C_M}$ and $\mathcal{H}^k \leq \frac{A_M\mu^2}{576C_M}$ for all $i \in [n]$ and $k \geq 0$. Then we have*

$$
\mathbb{E}[\Phi_4^k] \leq \theta_2^k \Phi_4^0,
$$

$$
\mathbb{E}\left[\frac{\|x^{k+1} - x^*\|^2}{\|x^k - x^*\|^2}\right] \leq \theta_2^k\left(\frac{3(L_*^2 + 4L_{\mathrm{F}}^2)A_M}{64C_W L_{\mathrm{F}}^2\mu^2} + \frac{12}{\mu^2}\right)\Phi_4^0.
$$

*for $k \geq 0$, where $\theta_2 := \left(1 - \frac{\min\{2A_W, A_M\}}{4}\right)$.*

*Proof.* Since $\xi^k \equiv 1$, $S^k \equiv [n]$, and $\mathcal{C}_{z_i^k, x^k}^M(x^{k+1}) \equiv x^{k+1}$ for all $i \in [n]$ and $k \geq 0$, we have $z_i^k \equiv w_i^k \equiv x^k$ for all $i \in [n]$ and $k \geq 0$. Then from (41), we have

$$
\mathbb{E}_k\|x^{k+1} - x^*\|^2 \leq \left(1 - \frac{3A_M}{8}\right)\|x^k - x^*\|^2. \tag{42}
$$

For $\|\mathbf{H}_i^{k+1} - \nabla^2 f_i(x^*)\|_{\mathrm{F}}^2$, similar to (36), we have

$$
\mathbb{E}_k\|\mathbf{H}_i^{k+1} - \nabla^2 f_i(x^*)\|_{\mathrm{F}}^2 \leq \left(1 - \frac{A_W}{2}\right)\|\mathbf{H}_i^k - \nabla^2 f_i(x^*)\|_{\mathrm{F}}^2 + C_W L_{\mathrm{F}}^2\|z_i^k - x^*\|^2 + C_W L_{\mathrm{F}}^2\mathbb{E}_k\|z_i^{k+1} - x^*\|^2.
$$

Considering $z_i^k \equiv x^k$, we further have

$$
\begin{aligned}
\mathbb{E}_k\|\mathbf{H}_i^{k+1} - \nabla^2 f_i(x^*)\|_{\mathrm{F}}^2 &\leq \left(1 - \frac{A_W}{2}\right)\|\mathbf{H}_i^k - \nabla^2 f_i(x^*)\|_{\mathrm{F}}^2 + C_W L_{\mathrm{F}}^2\|x^k - x^*\|^2 + C_W L_{\mathrm{F}}^2\mathbb{E}_k\|x^{k+1} - x^*\|^2 \\
&\overset{(42)}{\leq} \left(1 - \frac{A_W}{2}\right)\|\mathbf{H}_i^k - \nabla^2 f_i(x^*)\|_{\mathrm{F}}^2 + 2C_W L_{\mathrm{F}}^2\|x^k - x^*\|^2,
\end{aligned}
$$

which implies that

$$
\mathbb{E}_k[\mathcal{H}^{k+1}] \leq \left(1 - \frac{A_W}{2}\right)\mathcal{H}^k + 2C_W L_{\mathrm{F}}^2\|x^k - x^*\|^2. \tag{43}
$$

Thus, we have

$$
\begin{aligned}
\mathbb{E}_k[\Phi_4^{k+1}] &= \mathbb{E}_k[\mathcal{H}^{k+1}] + \frac{16C_W L_{\mathrm{F}}^2}{A_M} \mathbb{E}_k \|x^{k+1} - x^*\|^2 \\
&\leq \left(1 - \frac{A_W}{2}\right)\mathcal{H}^k + 2C_W L_{\mathrm{F}}^2 \|x^k - x^*\|^2 + \frac{16C_W L_{\mathrm{F}}^2}{A_M} \mathbb{E}_k \|x^{k+1} - x^*\|^2 \\
&\overset{(42)}{\leq} \left(1 - \frac{\min\{2A_W, A_M\}}{4}\right)\Phi_4^k.
\end{aligned}
$$

By applying the tower property, we have $\mathbb{E}[\Phi_4^{k+1}] \leq \theta_1 \mathbb{E}[\Phi_4^k]$. Unrolling the recursion, we have $\mathbb{E}[\Phi_4^k] \leq \theta_2^k \Phi_4^0$. Then we further have $\mathbb{E}[\mathcal{H}^k] \leq \theta_2^k \Phi_4^0$ and $\mathbb{E}\|x^k - x^*\|^2 \leq \frac{A_M}{16C_W L_{\mathrm{F}}^2} \theta_2^k \Phi_4^0$.

From (39), we can get

$$
\|x^{k+1} - x^*\|^2 \leq \frac{1}{\mu^2}\left(\frac{3(L_*^2 + 4L_{\mathrm{F}}^2)}{4}\|x^k - x^*\|^2 + 12\mathcal{H}^k\right)\|x^k - x^*\|^2.
$$

Assume $x^k \neq x^*$ for all $k \geq 0$. Then we have

$$
\frac{\|x^{k+1} - x^*\|^2}{\|x^k - x^*\|^2} \leq \frac{1}{\mu^2}\left(\frac{3(L_*^2 + 4L_{\mathrm{F}}^2)}{4}\|x^k - x^*\|^2 + 12\mathcal{H}^k\right),
$$

and by taking expectation, we arrive at

$$
\begin{aligned}
\mathbb{E}\left[\frac{\|x^{k+1} - x^*\|^2}{\|x^k - x^*\|^2}\right] &\leq \frac{3(L_*^2 + 4L_{\mathrm{F}}^2)}{4\mu^2}\mathbb{E}\|x^k - x^*\|^2 + \frac{12}{\mu^2}\mathbb{E}[\mathcal{H}^k] \\
&\leq \theta_2^k \left(\frac{3(L_*^2 + 4L_{\mathrm{F}}^2)A_M}{64C_W L_{\mathrm{F}}^2 \mu^2} + \frac{12}{\mu^2}\right)\Phi_4^0.
\end{aligned}
$$

$\square$

Next, we explore under what conditions we can guarantee the boundedness of $\|z_i^k - x^*\|^2$ and $\mathcal{H}^k$.

**Theorem H.3.** *Let Assumption 4.1 holds.*
*(i) Let $\mathcal{C}^M$ and $\mathcal{C}^W$ be deterministic. Assume $\|x^0 - x^*\|^2 \leq \frac{11A_M}{24C_M}\min\{\frac{A_M\mu^2}{36(L_*^2+4L_{\mathrm{F}}^2)C_M}, \frac{A_W A_M \mu^2}{2304 C_M C_W L_{\mathrm{F}}^2}\}$
and $\mathcal{H}^0 \leq \frac{A_M\mu^2}{576C_M}$. Then we have $\|x^k - x^*\| \leq \frac{11A_M}{24C_M}\min\{\frac{A_M\mu^2}{36(L_*^2+4L_{\mathrm{F}}^2)C_M}, \frac{A_W A_M \mu^2}{2304 C_M C_W L_{\mathrm{F}}^2}\}$, $\|z_i^k - x^*\|^2 \leq \min\{\frac{A_M\mu^2}{36(L_*^2+4L_{\mathrm{F}}^2)C_M}, \frac{A_W A_M \mu^2}{2304 C_M C_W L_{\mathrm{F}}^2}\}$ and $\mathcal{H}^k \leq \frac{A_M\mu^2}{576C_M}$ for all $i \in [n]$ and $k \geq 0$.*

*(ii) Assume $(z_i^k)_j$ is a convex combination of $\{(x^t)_j\}_{t=0}^k$, and $(\mathbf{H}_i^k)_{jl}$ is a convex combination of $\{(\nabla^2 f_i(z_i^k))_{jl}\}_{t=0}^k$ for all $i \in [n]$, $j, l \in [d]$, and $k \geq 0$. If $\|x^0 - x^*\|^2 \leq \tilde{c}_2 := \min\{\frac{2\mu^2}{3d^2(L_*^2+4L_{\mathrm{F}}^2)}, \frac{A_M\mu^2}{36dC_M(L_*^2+4L_{\mathrm{F}}^2)}, \frac{A_M\mu^2}{576d^3C_M L_\infty^2}, \frac{\mu^2}{24d^4L_\infty^2}\}$, then $\|z_i^k - x^*\|^2 \leq d\tilde{c}_2$ and $\mathcal{H}^k \leq \min\{\frac{A_M\mu^2}{576C_M}, \frac{\mu^2}{24d}\}$ for all $i \in [n]$ and $k \geq 0$.*

*Proof.* The proof is similar to that of Lemmas 5.2 and 5.3. Hence we omit it.

$\square$

# I Globalization Through Cubic Regularization and Line Search Procedure

So far, we have discussed only the local convergence of our methods. To prove global rates, one must incorporate additional regularization mechanisms. Otherwise, global convergence cannot be guaranteed. Due

to the smooth transition from contractive compressors to general 3PC mechanism, we can easily adapt two globalization strategies of FedNL (equivalent to Newton-EF21) to our Newton-3PC algorithm.

The two globalization strategies are *cubic regularization* and *line search procedure*. We only present the extension with cubic regularization Newton-3PC-CR (Algorithm 4) analogous to FedNL-CR (Safaryan et al., 2022). Similarly, line search procedure can be combined as it was done in FedNL-LS (Safaryan et al., 2022).

---

**Algorithm 4** Newton-3PC-CR (Newton's method with 3PC and Cubic Regularization)

---

1: **Input:** $x^0 \in \mathbb{R}^d$, $\mathbf{H}_1^0, \ldots, \mathbf{H}_n^0 \in \mathbb{R}^{d \times d}$, $\mathbf{H}^0 := \frac{1}{n} \sum_{i=1}^n \mathbf{H}_i^0$, $l^0 = \frac{1}{n} \sum_{i=1}^n \|\mathbf{H}_i^0 - \nabla^2 f_i(x^0)\|_{\mathrm{F}}$
2: **on** master
3:     $h^k = \arg \min_{h \in \mathbb{R}^d} T_k(h)$, where $T_k(h) := \langle \nabla f(x^k), h \rangle + \frac{1}{2} \langle (\mathbf{H}^k + l^k \mathbf{I}) h, h \rangle + \frac{L_*}{6} \|h\|^3$
4:     Update global model to $x^{k+1} = x^k + h^k$ and send to the nodes
5: **for** each device $i = 1, \ldots, n$ in parallel **do**
6:     Get $x^{k+1}$ and compute local gradient $\nabla f_i(x^{k+1})$ and local Hessian $\nabla^2 f_i(x^{k+1})$
7:     Take $\nabla^2 f_i(x^k)$ from memory and update $\mathbf{H}_i^{k+1} = \mathcal{C}_{\mathbf{H}_i^k, \nabla^2 f_i(x^k)}(\nabla^2 f_i(x^{k+1}))$
8:     Send $\nabla f_i(x^{k+1})$, $\mathbf{H}_i^{k+1}$ and $l_i^{k+1} := \|\mathbf{H}_i^{k+1} - \nabla^2 f_i(x^{k+1})\|_{\mathrm{F}}$ to the server
9: **end for**
10: **on** server
11:     Aggregate $\nabla f(x^{k+1}) = \frac{1}{n} \sum_{i=1}^n \nabla f_i(x^{k+1}), \mathbf{H}^{k+1} = \frac{1}{n} \sum_{i=1}^n \mathbf{H}_i^{k+1}, l^{k+1} = \frac{1}{n} \sum_{i=1}^n l_i^{k+1}$

---

We omit theoretical analysis of these extension as they can be obtained directly from FedNL approach with minor adaptations. In particular, one can get global linear rate for Newton-3PC-CR, global $\mathcal{O}(\frac{1}{k})$ rate for general convex case and the same fast local rates (9) and (11) of Newton-3PC.

## J  Additional Experiments and Extended Numerical Analysis

In this section we provide extended variety of experiments to analyze the empirical performance of Newton-3PC. We study the efficiency of Newton-3PC in different settings changing 3PC compressor and comparing with other second-order state-of-the-art algorithms. Tests were carried out on logistic regression problem with L2 regularization

$$\min_{x \in \mathbb{R}^d} \left\{ \frac{1}{n} \sum_{i=1}^n f_i(x) + \frac{\lambda}{2} \|x\|^2 \right\}, \quad f_i(x) = \frac{1}{m} \sum_{j=1}^m \log\left(1 + \exp(-b_{ij} a_{ij}^\top x)\right), \tag{44}$$

where $\{a_{ij}, b_{ij}\}_{j \in [m]}$ are data points at the $i$-th device. On top of that, we also consider L2 regularized Softmax problem of the form

$$\min_{x \in \mathbb{R}^d} \left\{ \frac{1}{n} \sum_{i=1}^n f_i(x) + \frac{\lambda}{2} \|x\|^2 \right\}, \quad f_i(x) = \sigma \log\left( \sum_{j=1}^m \exp\left( \frac{a_{ij}^\top x - b_{ij}}{\sigma} \right) \right), \tag{45}$$

where $\sigma > 0$ is a smoothing parameter. One can show that this function has both Lipschitz continuous gradient and Lipschitz continuous Hessian. Let $\tilde{a}_{ij}$ be initial data points, and $\tilde{f}_i$ be defined as

$$\tilde{f}_i(x) = \sigma \log\left( \sum_{j=1}^m \exp\left( \frac{\tilde{a}_{ij}^\top x - b_{ij}}{\sigma} \right) \right).$$

Then data shift is performed as follows

$$a_{ij} = \tilde{a}_{ij} - \tilde{f}_i(0), j \in [m], i \in [n].$$

After such shift we may claim that 0 is the optimum since $\nabla f(0) = 0$. Note that this problem does not belong to the class of *generalized linear models*.

### J.1 Datasets split

We use standard datasets from LibSVM library (Chang & Lin, 2011). We shuffle and split each dataset into $n$ equal parts representing a local data of $i$-th client. Exact names of datasets and values of $n$ are shown in Table 2.

**Table 2:** Datasets used in the experiments with the number of worker nodes $n$ used in each case.

| Data set | # workers $n$ | total # of data points $(= nm)$ | # features $d$ |
|---|---|---|---|
| a1a | 16 | 1600 | 123 |
| a9a | 80 | 32560 | 123 |
| w2a | 50 | 3450 | 300 |
| w8a | 142 | 49700 | 300 |
| phishing | 100 | 11000 | 68 |

### J.2 Choice of parameters

We follow the authors' choice of DINGO (Crane & Roosta, 2019) in choosing hyperparameters: $\theta = 10^{-4}, \phi = 10^{-6}, \rho = 10^{-4}$. Besides, DINGO uses a backtracking line search that selects the largest stepsize from $\{1, 2^{-1}, \ldots, 2^{-10}\}$. The initialization of $\mathbf{H}_i^0$ for Newton-3PC, FedNL (Safaryan et al., 2022) and its extensions, NL1 (Islamov et al., 2021) is $\nabla^2 f_i(x^0)$ if it is not specified directly. For Fib-IOS (Fabbro et al., 2022) we set $d_k^i = 1$. Local Hessians are computed following the partial sums of Fibonacci number and the parameter $\rho = \lambda_{q_{j+1}}$. This is stated in the description of the method. The parameters of backtracking line search for Fib-IOS are $\alpha = 0.5$ and $\beta = 0.9$.

We conduct experiments for two values of regularization parameter $\lambda \in \{10^{-3}, 10^{-4}\}$. In the figures we plot the relation of the optimality gap $f(x^k) - f(x^*)$ and the number of communicated bits per node. In the heatmaps numbers represent the communication complexity per client of Newton-3PC for some specific choice of 3PC compression mechanism (see the description in corresponding section). The optimal value $f(x^*)$ is chosen as the function value at the 20-th iterate of standard Newton's method.

In our experiments we use various compressors for the methods. Examples of classic compression mechanisms include Top-$K$ and Rank-$R$. The parameters of these compressors are parsed in details in Section A.3 of Safaryan et al. (2022); we refer a reader to this paper for disaggregated description of aforementioned compression mechanisms. Besides, we use various 3PC compressors introduced in (Richtárik et al., 2022).

### J.3 Behavior of Newton-CLAG based on Top-$K$ and Rank-$R$ compressors

Next, we study how the performance of Newton-CLAG changes when we vary parameters of biased compressor CLAG compression mechanism is based on. In particular, we test Newton-CLAG combined with Top-$K$ and Rank-$R$ compressors modifying compression level (parameters $K$ and $R$ respectively) and trigger parameter $\zeta$. We present the results as heatmaps in Figure 3 indicating the communication complexity in Mbytes for particular choice of a pair of parameters ($(K, \zeta)$ or $(R, \zeta)$ for CLAG based on Top-$K$ and Rank-$R$ respectively) .

First, we can highlight that in special cases Newton-CLAG reduces to FedNL ($\zeta = 0$, left column) and Newton-LAG (compression is identity, bottom row). Second, we observe slight improvement from using the lazy aggregation.

### J.4 Efficiency of Newton-3PCv2 under different compression levels

On the following step we study how Newton-3PCv2 behaves when the parameters of compressors 3PCv2 is based on are changing. In particular, in the first set of experiments we test the performance of Newton-3PCv2 assembled from Top-$K_1$ and Rand-$K_2$ compressors where $K_1 + K_2 = d$. Such constraint is forced to make the cost of one iteration to be $\mathcal{O}(d)$. In the second set of experiments we choose $K_1 = K_2 = K$ and vary $K$. The results are presented in Figure 4.

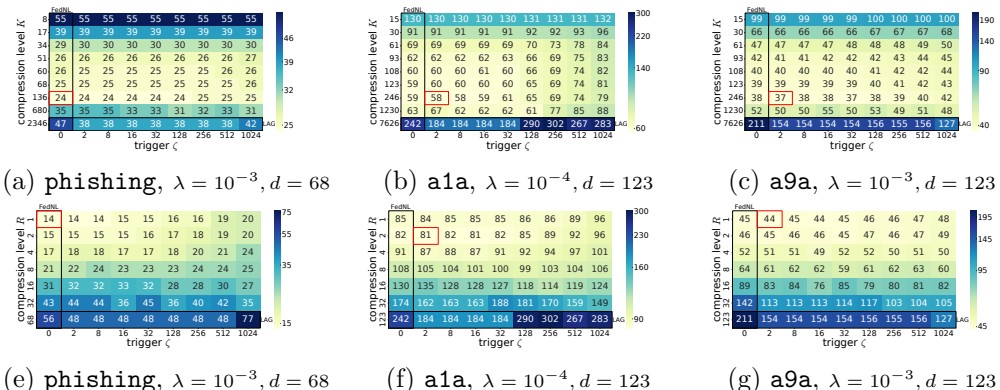

Figure 3: **First row:** The performance of Newton-CLAG based on Top-$K$ varying values of $(\zeta, K)$ in terms of communication complexity (in Mbytes). **Second row:** The performance of Newton-CLAG based on Rank-$R$ varying values of $(\zeta, R)$ in terms of communication complexity (in Mbytes).

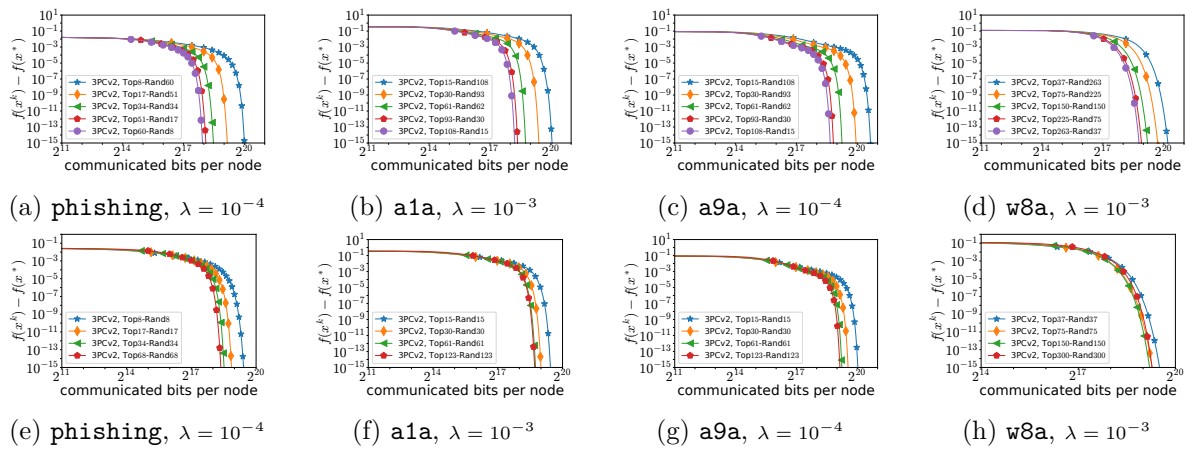

Figure 4: **First row:** The performance of Newton-3PCv2 where 3PCv2 compression mechanism is based on Top-$K_1$ and Rand-$K_2$ compressors with $K_1 + K_2 = d$ in terms of communication complexity. **Second row:** The performance of Newton-3PCv2 where 3PCv2 compression mechanism is based on Top-$K_1$ and Rand-$K_2$ compressors with $K_1 = K_2 \in \{d/8, d/4, d/2, d\}$ in terms of communication complexity.

For the first set of experiments, one can notice that randomness hurts the convergence since the larger the value of $K_2$, the worse the convergence in terms of communication complexity. In all cases a weaker level of randomness is preferable. For the second set of experiments, we observe that the larger $K$, the better communication complexity of Newton-3PCv2 except the case of w8a where the results for $K = 150$ are slightly better than those for $K = 300$.

### J.5 Behavior of Newton-3PCv4 **under different compression levels**

Now we test the behavior of Newton-3PCv4 where 3PCv4 is based on a pair (Top-$K_1$, Top-$K_2$) of compressors. Again, we have to sets of experiments: in the first one we examine the performance of Newton-3PCv4 when $K_1 + K_2 = d$; in the second one we check the efficiency of Newton-3PCv4 when $K_1 = K_2 = K$ varying $K$. In both cases we provide the behavior of Newton-EF21 (equivalent to FedNL) for comparison. All results are presented in Figure 5.

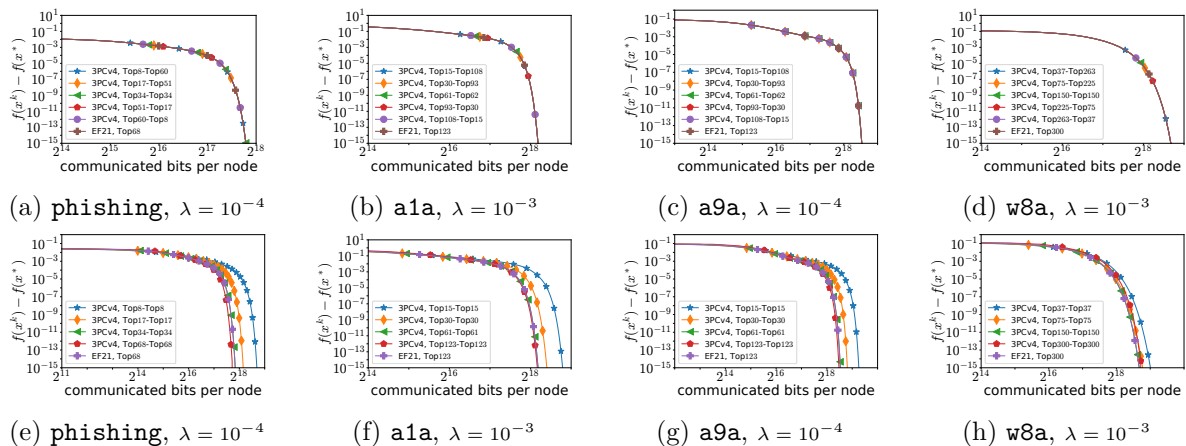

**Figure 5: First row:** The performance of Newton-3PCv4 where 3PCv4 compression mechanism is based on Top-$K_1$ and Top-$K_2$ compressors with $K_1 + K_2 = d$ in terms of communication complexity. **Second row:** The performance of Newton-3PCv4 where 3PCv4 compression mechanism is based on Top-$K_1$ and Top-$K_2$ compressors with $K_1 = K_2 \in \{d/8, d/4, d/2, d\}$ in terms of communication complexity. Performance of Newton-EF21 with Top-$d$ is given for comparison.

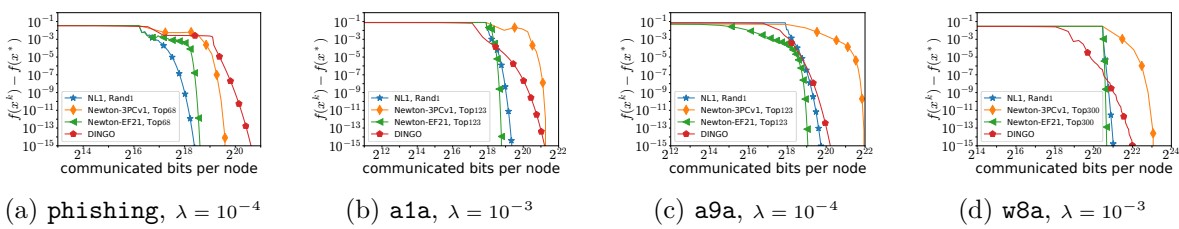

**Figure 6:** The performance of Newton-3PCv1 with 3PCv1 based on Top-$d$, Newton-EF21 (equivalent to FedNL) with Top-$d$, NL1 with Rand-1, and DINGO in terms of communication complexity.

As we can see, in the first set of experiments it does not matter how we distribute $d$ between $K_1$ and $K_2$ since it does not affect the performance. Regarding the second set of experiments, we can say that in some cases less aggressive compression ($K_1 = K_2 = d$) could be better than Newton-EF21.

### J.6 Study of Newton-3PCv1

Next, we investigate the performance of Newton-3PCv1 where 3PC compression mechanism is based on Top-$K$. We compare its performance with Newton-EF21 (equivalent to FedNL) with Top-$d$, NL1 with Rand-1, and DINGO. We observe in Figure 6 that Newton-3PCv1 is not efficient method since it fails in all cases.

### J.7 Performance of Newton-3PCv5

In this section we investigate the performance of Newton-3PCv5 where 3PC compression mechanism is based on Top-$K$. We compare its performance with Newton-EF21 (equivalent to FedNL) with Top-$d$, NL1 with Rand-1, and DINGO. According to the plots presented in Figure 7, we conclude that Newton-3PCv5 is not as effective as NL1 and Newton-EF21, but it is comparable with DINGO. The reason why Newton-3PCv5 is not efficient in terms of communication complexity is that we still need to send true Hessians with some nonzero probability which hurts the communication complexity of this method.

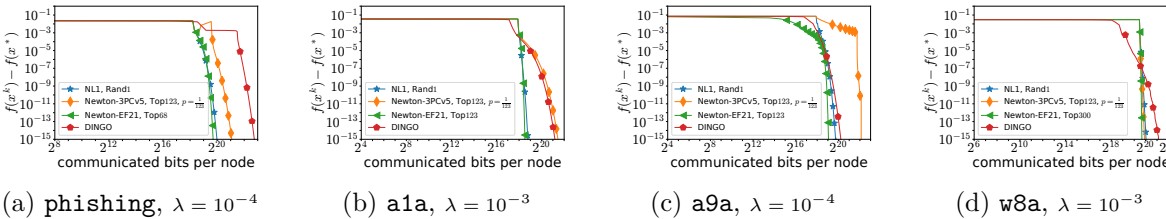

(a) phishing, $\lambda = 10^{-4}$     (b) a1a, $\lambda = 10^{-3}$     (c) a9a, $\lambda = 10^{-4}$     (d) w8a, $\lambda = 10^{-3}$

**Figure 7:** The performance of Newton-3PCv5 with 3PCv5 based on Top-$d$, Newton-EF21 (equivalent to FedNL) with Top-$d$, NL1 with Rand-1, and DINGO in terms of communication complexity.

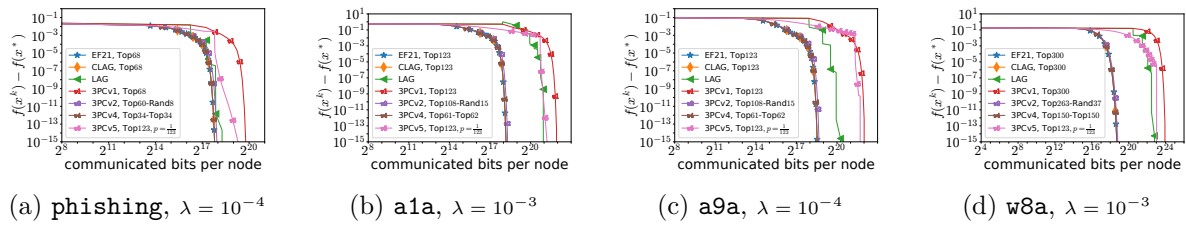

(a) phishing, $\lambda = 10^{-4}$     (b) a1a, $\lambda = 10^{-3}$     (c) a9a, $\lambda = 10^{-4}$     (d) w8a, $\lambda = 10^{-3}$

**Figure 8:** The performance of Newton-3PC with different choice of 3PC compression mechanism in terms of communication complexity.

### J.8   Newton-3PC **with different choice of 3PC compression mechanism**

Now we investigate how the choice of 3PC compressor influences the communication complexity of Newton-3PC. We test the performance of Newton-3PC with EF21, CLAG, LAG, 3PCv1 (based on Top-$K$), 3PCv2 (based on Top-$K_1$ and Rand-$K_2$), 3PCv4 (based on Top-$K_1$ and Top-$K_2$), and 3PCv5 (based on Top-$K$). We choose $p = 1/d$ for Newton-3PCv5 in order to make the communication cost of one iteration to be $\mathcal{O}(d)$. The choice of $K$, $K_1$, and $K_2$ is justified by the same logic.

We clearly see that Newton-3PC combined with EF21 (Newton-3PC with this 3PC compressor reduces to FedNL), CLAG, 3PCv2, 3PCv4 demonstrates almost identical results in terms of communication complexity. Newton-LAG performs worse than previous methods except the case of phishing dataset. Surprisingly, Newton-3PCv1, where only true Hessian differences is compressed, demonstrates the worst performance among all 3PC compression mechanisms. This probably caused by the fact that communication cost of one iteration of Newton-3PCv1 is significantly larger than those of other Newton-3PC methods.

### J.9   **Analysis of Bidirectional** Newton-3PC

#### J.9.1   **EF21 compression mechanism**

In this section we analyze how each type of compression (Hessians, iterates, and gradients) affects the performance of Newton-3PC. In particular, we choose Newton-EF21 (equivalent to FedNL) and change parameters of each compression mechanism. For Hessians and iterates we use Top-$K_1$ and Top-$K_2$ compressors respectively. In Figure 9 we present the results when we vary the parameter $K_1, K_2$ of Top-$K$ compressor and probability $p$ of Bernoulli Aggregation. The results are presented as heatmaps indicating the number of Mbytes transmitted in uplink and downlink directions by each client.

In the first row in Figure 9 we test different combinations of compression parameters for Hessians and iterates keeping the probability $p$ of BAG for gradients to be equal 0.5. In the second row we analyze various combinations of pairs of parameters $(K, p)$ for Hessians and gradients when the compression on iterates is not applied. Finally, the third row corresponds to the case when Hessians compression is fixed (we use Top-$d$), and we vary pairs of parameters $(K, p)$ for iterates and gradients compression.

According to the results in the heatmaps, we can conclude that Newton-EF21 benefits from the iterates compression. Indeed, in both cases (when we vary compression level applied on Hessians or gradients) the best result is given in the case when we do apply the compression on iterates. This is not the case for gradients (see second row) since the best results are given for high probability $p$; usually for $p = 1$ and rarely for $p = 0.75$. Nevertheless, we clearly see that bidirectional compression is indeed useful in almost all cases.

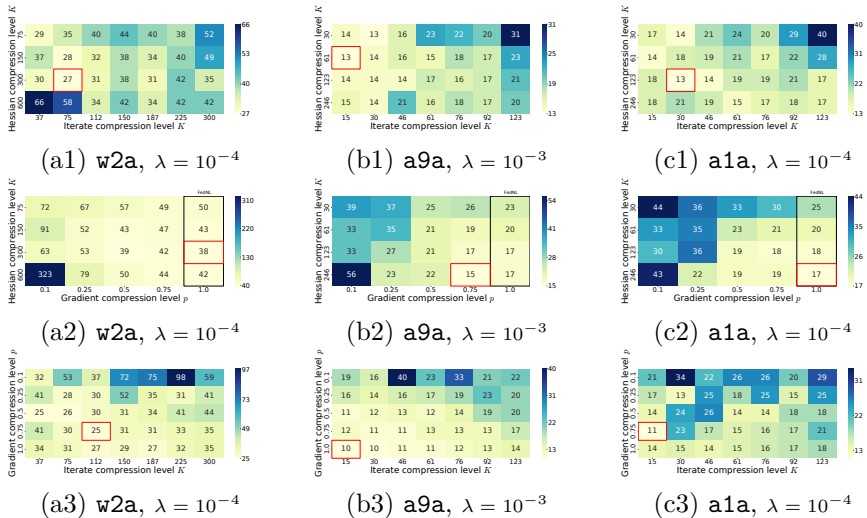

| (a1) w2a, $\lambda = 10^{-4}$ | (b1) a9a, $\lambda = 10^{-3}$ | (c1) a1a, $\lambda = 10^{-4}$ |
|---|---|---|
| (a2) w2a, $\lambda = 10^{-4}$ | (b2) a9a, $\lambda = 10^{-3}$ | (c2) a1a, $\lambda = 10^{-4}$ |
| (a3) w2a, $\lambda = 10^{-4}$ | (b3) a9a, $\lambda = 10^{-3}$ | (c3) a1a, $\lambda = 10^{-4}$ |

**Figure 9: First row:** The performance of Newton-3PC-BC in terms of communication complexity for different values of $(K_1, K_2)$ of Top-$K_1$ and Top-$K_2$ compressors applied on Hessians and iterates respectively while probability $p = 0.75$ of BAG applied on gradients is fixed. **Second row:** The performance of Newton-EF21 in terms of communication complexity for different values of $(K_1, p)$ of Top-$K_1$ compressor applied on Hessians and probability $p$ of BAG applied on gradients while $K_2 = d$ parameter of Top-$K_2$ applied on iterates is fixed. **Third row:** The performance of Newton-EF21 in terms of communication complexity for different values of $(K_2, p)$ of Top-$K_2$ compressor applied on iterates and probability $p$ of BAG applied on gradients while $K_1 = d$ parameter of Top-$K_1$ applied on Hessians is fixed.

### J.9.2 3PCv4 compression mechanism

In our next set of experiments we fix EF21 compression mechanism based on Top-$d$ compressor applied on Hessians and probability $p = 0.75$ of Bernoulli aggregation applied on gradients. Now we use 3PCv4 update rule on iterates based on outer and inner compressors (Top-$K_1$, Top-$K_2$) varying the values of pairs $(K_1, K_2)$. We report the results as heatmaps in Figure 10.

We observe that in all cases it is better to apply relatively smaller outer and inner compression levels as this leads to better performance in terms of communication complexity. Note that the first row in heatmaps corresponds to Newton-3PC-BC when we apply just EF21 update rule on iterates. As a consequence, Newton-3PC-BC reduces to FedNL-BC method (Safaryan et al., 2022). We obtain that 3PCv4 compression mechanism applied on iterates in this setting is more communication efficient than EF21. This implies the fact that Newton-3PC-BC could be more efficient than FedNL-BC in terms of communication complexity.

### J.10 BL1 (Qian et al., 2022) with 3PC compressor

As it was stated in Section 4.1 Newton-3PC covers methods introduced in (Qian et al., 2022) as a special case. Indeed, in order to run, for example, BL1 method we need to use rotation compression operator 20. The role of orthogonal matrix in the definition plays the basis matrix.

In this section we test the performance of BL1 in terms of communication complexity with different 3PC compressors: EF21, CBAG, CLAG. For CBAG update rule the probability $p = 0.5$, and for CLAG the

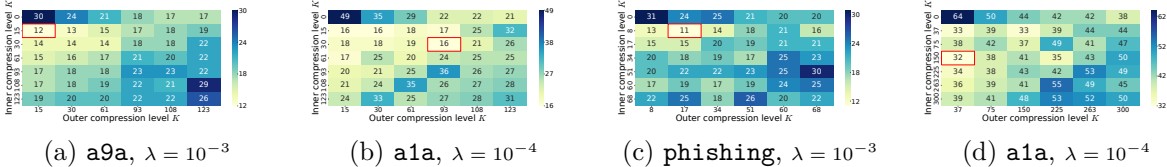

**Figure 10:** The performance of Newton-3PC-BC with EF21 update rule based on Top-$d$ compressor applied on Hessians, BAG update rule with probability $p = 0.75$ applied on gradients, and 3PCv4 update rule based on (Top-$K_1$, Top-$K_2$) compressors applied on iterates for different values of pairs $(K_1, K_2)$.

trigger $\zeta = 2$. All aforementioned 3PC compression operators are based on Top-$\tau$ compressor where $\tau$ is the dimension of local data (see Section 2.3 of (Qian et al., 2022) for detailed description).

Observing the results in Figure 11, we can notice that there is no improvement of one update rule over another. However, in EF21 is slightly better than other 3PC compressors in a half of the cases, and CBAG insignificantly outperform in other cases. This means that even if the performance of BL1 with EF21 and CBAG are almost identical, CBAG is still preferable since it is computationally less expensive since we do not need to compute local Hessians and their representations in new basis.

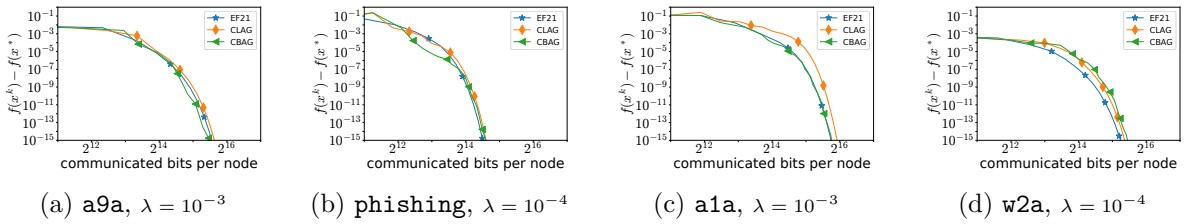

**Figure 11:** The performance of BL1 with EF21, CBAG and CLAG 3PC compression mechanisms in terms of communication complexity.

### J.11 Analysis of Newton-3PC-BC-PP

#### J.11.1 3PC's parameters fine-tuning for Newton-3PC-BC-PP

On the following step we study how the choice of parameters of 3PC compression mechanism and the number of active clients influence the performance of Newton-3PC-BC-PP.

In the first series of experiments we test Newton-3PC-BC-PP with CBAG compression combined with Top-$2d$ compressor and probability $p$ applied on Hessians; EF21 with Top-$2d/3$ compressor applied on iterates; BAG update rule with probability $p = 0.75$ applied on gradients. We vary aggregation probability $p$ of Hessians and the number of active clients $\tau$. Looking at the numerical results in Figure 12 (first row), we may claim that the more clients are involved in the optimization process in each communication round, the faster the convergence since the best results in each case always belongs the first column. However, we do observe that lazy aggregation rule with probability $p < 1$ is still beneficial.

In the second row of Figure 12 we investigate Newton-3PC-BC-PP with CBAG compression based on Top-$d$ and probability $p = 0.75$ applied on Hessians; 3PCv5 update rule combined with Top-$2d/3$ and probability $p$ applied on iterates; BAG lazy aggregation rule with probability $p - 0.75$ applied gradients. In this case we modify iterate aggregation probability $p$ and the number of clients participating in the training. We observe that again the fastest convergence is demonstrated when all clients are active, but aggregation parameter $p$ of iterates smaller than 1.

Finally, we study the effect of BAG update rule on the communication complexity of Newton-3PC-BC-PP. As in previous cases, Newton-3PC-BC-PP is more efficient when all clients participate in the training process.

Nevertheless, lazy aggregation rule of BAG still brings the benefit to communication complexity of the method.

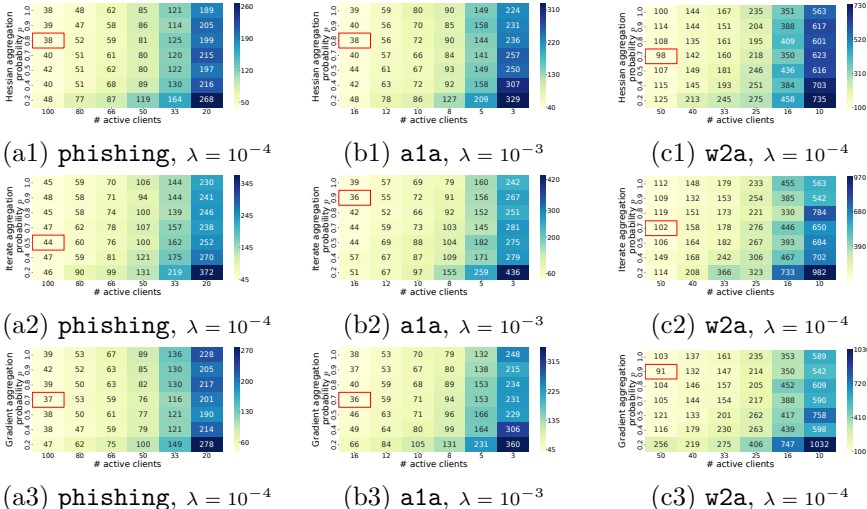

**Figure 12:** The performance of Newton-3PC-BC-PP with various update strategies in terms of communication complexity (in Mbytes).

### J.11.2 Comparison of different 3PC update rules

Now we test different combinations of 3PC compression mechanisms applied on Hessians and iterates. First, we fix probability parameter of BAG update rule applied on gradients to $p = 0.7$. The number of active clients in all cases $\tau = {}^n\!/_2$. We analyze various combinations of 3PC compressors: CBAG (Top-$d$ and $p = 0.7$) and 3PCv5 (Top-${}^d\!/_2$ and $p = 0.7$); EF21 (Top-$d$) and EF21 (Top-${}^d\!/_2$); CBAG (Top-$d$ and $p = 0.7$) and EF21 (Top-${}^d\!/_2$); EF21 (Top-$d$) and 3PCv5 (Top-${}^d\!/_2$ and $p = 0.7$) applied on Hessians and iterates respectively. Numerical results might be found in Figure 13. We can see that in all cases Newton-3PC-BC-PP performs the best with a combination of 3PC compressors that differ from EF21+EF21. This allows to conclude that EF21 update rule is not always the most effective since other 3PC compression mechanisms lead to better performance in terms of communication complexity. Nonetheless one can notice that it is useless to apply CBAG or LAG compression mechanisms on iterates. Indeed, in the case when we skip communication the iterates remain intact, and the next step is equivalent to previous one. Thus, there is no need to carry out the step again.

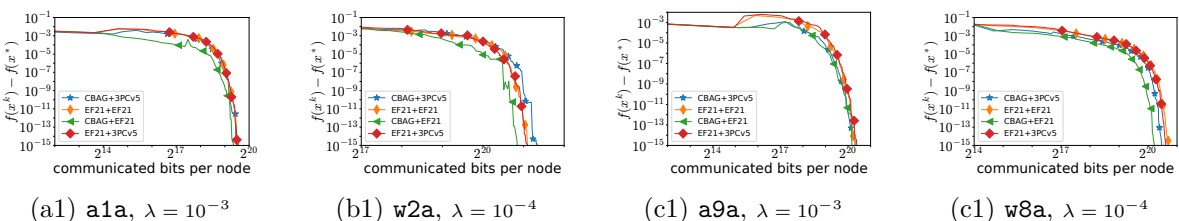

**Figure 13:** The performance of Newton-3PC-BC-PP with different combinations of 3PC compressors applied on Hessians and iterates respectively.

### J.12 Global convergence of Newton-3PC

Now we investigate the performance of globally convergent Newton-3PC-LS — an extension of Newton-3PC — based on the line search as it performs significantly better than Newton-3PC-CR based on cubic

regularization. The experiments are done on synthetically generated datasets with heterogeneity control. A detailed description of how the datasets are created is given in section B.12 of (Safaryan et al., 2022). Roughly speaking, the generation function has 2 parameters $\alpha$ and $\beta$ that control the heterogeneity of local data. We denote datasets created in a such way with parameters $\alpha$ and $\beta$ as `Synt(`$\alpha$`, `$\beta$`)`. All datasets are generated with dimension $d = 100$, split between $n = 20$ clients each of which has $m = 1000$ local data points. In all cases the regularization parameter is chosen $\lambda = 10^{-4}$.

We compare 5 versions of Newton-3PC-LS combined with EF21 (based on Rank-1 compressor), CBAG (based on Rank-1 compressor with probability 0.8), CLAG (based on Rank-1 compressor and communication trigger $\zeta = 2$), 3PCv2 (based on Top-$^{3d}/_4$ and Rand-$^{d}/_4$ compressors), and 3PCv4 (based on Top-$^{d}/_2$ and Top-$^{d}/_2$ compressors). In this series of experiments the initialization of $\mathbf{H}_i^0$ is equal to zero matrix. The comparison is performed against ADIANA (Li et al., 2020b) with random dithering ($s = \sqrt{d}$), Fib-IOS (Fabbro et al., 2022), and GIANT (Wang et al., 2018).

The numerical results are shown in Figure 14. According to them, we observe that Newton-3PC-LS is more resistant to heterogeneity than other methods since they outperform others *by several orders in magnitude*. Besides, we see that Newton-CBAG-LS and Newton-EF21-LS are the most efficient among all Newton-3PC-LS methods; in some cases, the difference is considerable.

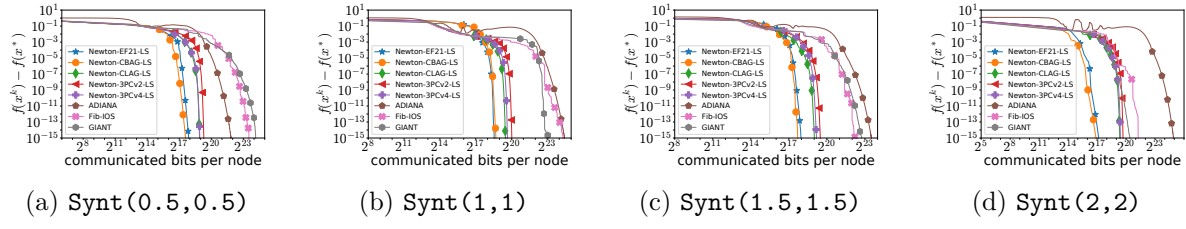

(a) `Synt(0.5,0.5)`     (b) `Synt(1,1)`     (c) `Synt(1.5,1.5)`     (d) `Synt(2,2)`

**Figure 14:** The performance of Newton-3PC-LS with different combinations of 3PC compressors applied on Hessians against ADIANA, Fib-IOS, and GIANT.

