# OpenReview forum: "Distributed Newton-Type Methods with Communication Compression and Bernoulli Aggregation"
_TMLR — Accepted by TMLR_

### Review · Reviewer_tB6q · 2023-07-30

**Summary Of Contributions:**

The paper proposes the use of 3pc compressors for a few federated learning settings.

**Audience:**

Yes

**Claims And Evidence:**

Yes

**Requested Changes:**

Please see the strenghts/weaknesses section.

**Strengths And Weaknesses:**

Strengths:
The empirical results are promising and adequate. The presented algorithms and theory and clear to read and understand. I have not gone through all the proofs in the

Questions/weaknesses:
The paper rehashes known techniques, so novelty is limited but that is not what tmlr is about.

I would suggest add more information about aggregation mechanisms and what that implies, since it is an equal part of the CLAG mechanisms.

 Some of the “lemmas” such as 3.6 and 3.4 are trivial and could be written as sentences to save place, if needed.

Generally the convergence rates are known where the shrinkage factor (e.g. \rho in Theorem 4.2) depends on the condition number. Even if there is no compression, should the convergence rate not “fall back”to what is known about strongly convex functions? It would be nice if the authors can provide some intuition on why this does not happen, and why the presented rates turn out to be free of the condition number.

The locality conditions mentioned in Lemmas 4.3 and 4.4 are required to hold for \mathcal{H^k} for Theorem 4.2, and not really required for || x^k – x^*||^2, no ? why is the latter also mentioned in the lemmas, when it can be skipped and it is not useful ?

This is not really related to the paper itself or its contributions, but are the authors aware of approximate compression techniques that does not required full instantiation/calculation of the local hessians? The use of sketch and project operator is unclear, and it is also not clear how to hunt for "good" compression operators, since the analysis itself assumes this is known already.

---

> ### Author Response · Authors · 2023-08-03
> **Response to Reviewer tB6q**
>
> We would like to thank the reviewer for his/her valuable comments and suggestions! Let us clarify some of the points you mentioned.
>
> $\textbf{Q: The paper rehashes known techniques, so novelty is limited but that is not what tmlr is about.}$
>
> $\textbf{A:}$ The compression mechanisms are well studied for gradient-type algorithms. There are many papers showing advantageous in different settings. However, this is completely not the case for second-order methods. Typically papers use techniques inherited from first-order literature and show their practical benefits while the theoretical part is missing or bad (e.g., the rates are worse than those of first-order methods). In our work we do show that wide range of compression techniques provably work when applied to second-order methods. Together with known compression mechanisms we also introduce new once which are covered by our theory and demonstrate good practical performance.
> For instance, one of the compression techniques we propose allows skipping both computations and communication of the Hessian periodically.
>
> $ \textbf{Q: I would suggest add more information about aggregation mechanisms and what that implies, since it is an equal part of the CLAG mechanisms.}$
>
> $\textbf{A:}$ Thank you for the suggestion. Below we post the possible discussion on lazy aggregation meschanisms that can be added to the main part.
>
> Lazy aggregation communication mechanisms empirically outperform Gradient Descent [1]. The main idea is to communicate local gradients/Hessians only in the case when a certain conidition holds. For example, in the case of CLAG we update local Hessian estimators only if they are far from true local Hessians in L2 norm. Parameter $\zeta$ controls how often we want to skip communication. If it is large, then we reuse previous Hessian estimators more often. CLAG can be seen as an adaptive mechanism that dynaimcally updates estimators based on the conditions that change throughout the optimization process. Thus, it is difficult to estimate/approximate how frequently we skip communications.
>
> In fact, lazy aggregation remains poorly studied field in the literature. It has been analyzed for gradient-type methods only. However, the theoretical analysis show either explicit convergence guarantees [1, 2], or sublinear rates in the strongly convex regime [3]. Recently, [4] have shown that LAG can be properly studied compression point of view [4]. They derive optimal convergence guarantees suported by emprirical evaluations. The follow-up work [5] uses similar idea in one node regime. They compute Hessian deterministically once in 1/d iterations. They theoretical analysis also show benefits of exploiting rare updates. Nevertheless, the lazy aggregation in the case of second-order methods remains poorly studied. In our work we make a step towards better understanding of lazy aggregation for Newton-type algorithms.
>
>
> [1] Chen, T., Giannakis, G., Sun, T., and Yin, W. LAG: Lazily aggregated gradient for communication-efficient distributed learning. Advances in Neural Information Processing Systems, 2018.
>
> [2] Sun, J., Chen, T., Giannakis, G., and Yang, Z. Communication-efficient distributed learning via lazily
> aggregated quantized gradients. Advances in Neural Information Processing Systems, 32:3370–3380, 2019.
>
> [3] Ghadikolaei, H. S., Stich, S., and Jaggi, M. LENA: Communication-efficient distributed learning with self- triggered gradient uploads. In International Conference on Artificial Intelligence and Statistics, pp. 3943–3951. PMLR, 2021.
>
> [4] Peter Richtarik, Igor Sokolov, Ilyas Fatkhullin, Elnur Gasanov, Zhize Li, and Eduard Gorbunov. 3pc: Three point compressors for communication-efficient distributed training and a better theory for lazy aggregation. 39th International Conference on Machine Learning, 2022.
>
> [5] Nikita Doikov, El Mahdi Chayti, Martin Jaggi, Second-order optimization with lazy Hessians, ICML 2023.
>
> $\textbf{Q: Some of the “lemmas” such as 3.6 and 3.4 are trivial and could be written as sentences to save place, if needed.} $
>
> $\textbf{A:}$ First, we do not seem to have issues with saving space, the main part of the paper is not long and converting those lemmas into sentences will not save much space neither. Second, if we replace those lemmas into statements, then it will be harder for the reader to link them with their corresponding proofs in the appendix.

---

> ### Author Response · Authors · 2023-08-03
> **Response to Reviewer t6Bq (Part 2)**
>
> $\textbf{Q: Generally the convergence rates are known where the shrinkage factor (e.g. $\rho$ in Theorem 4.2) depends on the condition number.}$ $\textbf{Even if there is no compression, should the convergence rate not “fall back”to what is known about strongly convex functions?}$ $\textbf{It would be nice if the authors can provide some intuition on why this does not happen, and why the presented rates turn out to be free of}$ $\textbf{the condition number.}$
>
> $\textbf{A:}$ The shrinkage factor $\rho$ depends on the condition number for any first-order algorithm. However, for the properly designed second-order methods, in particular the classical Newton's method, the shrinkage factor does  not depend on problem conditioning. For instance, the convergence of the classical Newton's method is locally quadratic and is given by the inequality $\|x_{k+1} - x^\star \| \leq C\|x_{k} - x^\star \|^2$ where $C$ is some constant. Now if we focus on local convergence, namely initialize $x_0$ in such a way that $\|x_0 - x^{\star}\| \le \frac{1}{2C}$, then it is trivial to notice that $\|x_{k+1}-x^{\star} \| \leq \frac{1}{2}\|x_k - x^{\star}\|$ for all $k\ge0$. As we can see, the shrinkage factor $\frac{1}{2}$ does not depend on conditioning of the problem since the method utilizes local curvature information (Hessian matrices) in each iteration. Our proposed Newton-3PC method can be seen as an approximation of the classical Newton's method, and as such inherits condition-independent property of the convergence rate.
>
> $\textbf{Q: This is not really related to the paper itself or its contributions, but are the authors aware of approximate compression techniques that does not}$ $\textbf{required full instantiation/calculation of the local hessians?}$ $\textbf{The use of sketch and project operator is unclear and it is also not clear }$, $\textbf{how to hunt for "good" compression operators, since the analysis itself assumes this is known already.}$
>
> $\textbf{A:}$ Every approach we know that allows to use compression and avoid full Hessian computation involves sketching in one or another way since we always have to compute second order derivatives along some direction, i.e. rely on Hessian-vector products. That is why we present sketch and project operator. Now we want to clarify the use of sketch and project operator.
>
> It saves computation time: indeed, in order to compute the output of sketch and project operator we need to compute Hessian-vector products (e.g., Hessian times i-th unit basis vector requires computing the i-th column of Hessian only).
>
> It saves communication time: if we use sketching with small $\tau$ (e.g., $\tau = O(1)$) when sketch and project is a compression operator as well since we need to broadcast $d\tau = O(d)$ floats to the server.
>
> It is supported by the theory: according to lemma 3.5 sketch and project operator is contractive with certain contraction factor. This means that we can provably use it within Newton-3PC framework.
>
> Practically known that contractive compressors outperform unbiased compressors. For example, compressors such as Top-K and Rank-R are among the most popular and used ones for second-order algorithms; see Figure~1 (first row). Besides, recent results [6] show that Top-K in combination with EF21 (this combination is a special case of the proposed class of 3PC compressors) provably improves the performance when features have some sparsity patterns. [7] show that rotation compressors (see the definition Appendix, Section E.5) in combination with Top-k or Rank-R improves further the performance of the algorithm if local data belongs to low dimensional space. Finally, a compressor can be chosen through grid search as well.
>
> [6] Peter Richtárik, Elnur Gasanov, Konstantin Burlachenko, Error Feedback Shines when Features are Rare, arXiv preprint arXiv:2305.15264
>
> [7] Xun Qian, Rustem Islamov, Mher Safaryan, Peter Richtárik. Basis Matters: Better Communication-Efficient Second Order Methods for Federated Learning, AISTATS 2022

---

### Review · Reviewer_22SC · 2023-08-12

**Summary Of Contributions:**

This paper stuides distributed Newton method with communication compression. Specifically, the three point compressor (3PC) of Richtarik et al. [2022] is combined with the traditional Newton method. Several new 3PC mechanisms, such as adaptive thresholding and Bernoulli aggregation, are discovered in this paper. Fast condition-number-independent local linear and/or superlinear convergence rates are derived for the proposed methods.

**Audience:**

Yes

**Claims And Evidence:**

Yes

**Requested Changes:**

1. I suggest the authors to discuss the challenges in the proof of Theorem 4.2, especially something new beyong the proofs in the Newton method and 3PC.

2. I suggest the authors to compare with the uncompressed distributed Newton method in the first row of Figure 1, which communicates the full Hessian matrix. Then, the readers will know how many bits are saved after compression.

3. In Table 1, the communication cost is O(d). However, the proposed methods need to send the compressed Hessian matrix to the server. Is this because the compresser only sets the top-d elements non-zero in the d*d matrix?

**Strengths And Weaknesses:**

Strengths:

1. The proposed Newton-3PC method is clean.

2. The theory is solid with local linear and/or superlinear convergence rates.

Weaknesses:

1. The improvement of the proposed methods over the SOTA methods is not significant. In the first row of Figure 1, to achieve the same accuracy, the proposed method only saves half communication bits compared with NL1 or DINGO.

2. I am not sure if it is a trival combination of 3PC and Newton method. I suggest the authors to discuss the challenges in the proof of Theorem 4.2, especially something new beyong the proofs in the Newton method and 3PC.

---

### Review · Reviewer_pEt4 · 2023-08-27

**Summary Of Contributions:**

This paper discusses Newton-type convex optimization methods in a distributed setting. Specifically, the authors consider a general approach using three point compressors for communicating Hessian information. A variety of compressors and aggregation schemes are explored, and two new techniques for deciding when to aggregate are proposed. Local linear and superlinear convergence rates are established for this general method depending on the quantity under consideration.

**Audience:**

Yes

**Broader Impact Concerns:**

No ethical concerns.

**Claims And Evidence:**

Yes

**Requested Changes:**

Overall, the paper needs to be thoroughly updated to provide a clearer presentation of the author's work and a more focused  discussion of their specific contributions. Addressing the weaknesses given above provide a starting point for these changes.

**Strengths And Weaknesses:**

## Strengths
When viewed as a generalization of FedNL to arbitrary three point compressors, while maintaining convergence guarantees, the core idea seems strong. As well, the idea of randomized updating through Bernoulli aggregation has clear benefits in terms of reducing complexity.

## Weaknesses
Overall, the paper suffers from a general lack of clarity in exposition and focus on what makes this work novel.

-  One of the main suggested contributions is the extension of three point compressors to Hessian communication in Definition 3.1. However, using the Frobenius norm for matrices is equivalent to using the 2-norm for vectors, so the original definition suffices if the matrix is simply reshaped. Thus the novelty of this purported extension is questionable.
- Fast local convergence rates are the focus of the theoretical analysis, yet in many cases it seems this would likely not be the dominant cost. Globalization strategies are partially presented in the appendix, but there is limited discussion on the subject in the main text.
- In Section 4.2 it is stated that "The key novelty Newton-3PC brings" is in reducing communication and computational complexity. While the communication complexity depends on the compressor being used, it seems that the computational complexity on any local node is still large as a new Hessian must be computed and stored at every step even if there is no update.
- Lemma 4.4 appears to have no dependence on the probability of updating the estimate, which begs the question of what happens when $p=0$, i.e. no update is ever sent?
- In Section 4.4 local convergence rates are provided for three different quantities, yet discussion surrounding these results is lacking.
- In Section 6.2 and Figure 1 the benefits of Bernoulli aggregation on communcation and computational complexity are discussed, but these results seem somewhat trivial. Infrequent updates necessarily result in a reduction of costs.
- Unclear algorithm presentation. The following specifically apply to Algorithm 1, but Algorithm 2 has similar issues.
	- How are the initial local Hessian estimates computed?
	- The option to send the local error seems unnecessarily confusing, perhaps a comment that this step is only necessary if using the second update strategy would suffice.
	- Line 9 always sends the local updated local Hessian estimate to the server, but this will always incur a $d^2$ communication cost even if there has been no update
- Unclear language
	- In the discussion of Example 3.3, the idea of Top-K is referenced but not defined. This, and other compressors such as Rank-K are again brought up in Section 6, but never clearly defined.
	- The second paragraph of Section 5 introduces three sets of parameters, but where these parameters live and their purpose is not discussed.
	- Throughout Section 3 the domain and range are given as $\mathrm{R}^d$ when it should be $\mathrm{R}^{d\times d}$.

---

### Decision · Action_Editors · 2023-10-03

**Recommendation:** Accept as is

**Comment:**

All three reviewers agreed there were interesting ideas, and that the paper was by-and-large well-executed; the paper is also a good length, which may help its impact. In terms of clarity, the high-level ideas were clear, but even after the rebuttal and revision, reviewer pEt4 still felt the details of the presentation could be cleaned up to make the paper more clear.  If the authors wish to make any minor changes related to clarity before submitting the camera-ready version, that is fine with me [also for the camera-ready version, if the authors have github code that they did not include a link to before since it would de-anonymize the paper, they may choose to add it back in].

**Audience:**

Fast optimization methods obviously have a very broad audience. This paper considers the subtopic of distributed optimization, which is increasingly relevant (e.g., federated learning) as well.  Overall, I think this paper has a larger-than-average audience for TMLR papers.

**Claims And Evidence:**

The paper provides both theory and experiments. The main text mostly centers on local convergence theory, demonstrating linear convergence with a rate that is independent of the condition number (hence a clear improvement [locally] over first-order methods), although there is also discussion of globalization via cubic regularization + linesearch in the appendix.  There are significant experiments that also illustrate points on the communication cost.

None of the reviewers raised any serious concern over the correctness of the results, and I don't see any issues myself.  As is typical, not every line of the proof was checked, but there were no "suspicious" claims that caught the reviewers' attention.